# Differentiation granules, a dynamic regulator of *T. brucei* development

**Mathieu Cayla** [1,2] ✉, **Christos Spanos** [3], **Kirsty McWilliam** [1], **Eliza Waskett**[1], **Juri Rappsilber**[3] & **Keith R. Matthews** [1]

Adaptation to a change of environment is an essential process for survival, in particular for parasitic organisms exposed to a wide range of hosts. Such adaptations include rapid control of gene expression through the formation of membraneless organelles composed of poly-A RNA and proteins. The African trypanosome *Trypanosoma brucei* is exquisitely sensitive to well-defined environmental stimuli that trigger cellular adaptations through differentiation events that characterise its complex life cycle. The parasite has been shown to form stress granules in vitro, and it has been proposed that such a stress response could have been repurposed to enable differentiation and facilitate parasite transmission. Therefore, we explored the composition and positional dynamics of membraneless granules formed in response to starvation stress and during differentiation in the mammalian host between the replicative slender and transmission-adapted stumpy forms. We find that *T. brucei* differentiation does not reflect the default response to environmental stress. Instead, the developmental response of the parasites involves a specific and programmed hierarchy of membraneless granule assembly, with distinct components and regulation by protein kinases such as TbDYRK, that are required for the parasite to successfully progress through its life cycle development and prepare for transmission.

African trypanosomes represent an excellent model for the understanding of the posttranscriptional regulation of gene expression in a eukaryotic system. These parasites, responsible for human diseases as well as lost livestock productivity in the tsetse belt in Africa, undergo a number of well-characterised developmental transitions as they progress through their life cycle in both a mammalian host and in their disease vector, the tsetse fly. Gene regulation in these parasites is achieved predominantly through the control of mRNA stability and translation, with transcriptional control relatively unimportant due to the polycistronic organisation of the parasite's genome and its paucity of gene specific promoters. Consequently, as parasites progress between life cycle stages, or respond to specific environmental cues, gene regulation is achieved by the regulated turnover or translation of

specific mRNAs. Where this involves an abrupt transition between distinct environments (as with the transmission between the mammalian host and insect vector) or occurs under conditions of nutritional stress (such as glucose depletion), rapid changes are required that generate both mRNA and protein changes necessary to ensure adaptation and survival.

Analysis of the extensive regulation of the trypanosome's gene expression profile under different environmental conditions has resulted in the identification and characterisation of distinct mRNA regulators. Some of these can individually activate quite extensive phenotypic changes in the parasite, an example being RBP6, which can drive progression through several life cycle stages without a physiological environmental stimulus[1,2]. However, in other cases an

[1]Institute for Immunology and Infection Research, School of Biological Sciences, University of Edinburgh, Edinburgh, UK. [2]York Biomedical Research Institute, Department of Biology, University of York, York, UK. [3]Wellcome Centre for Cell Biology, University of Edinburgh, Edinburgh, UK. ✉e-mail: mathieu.cayla@york.ac.uk

involvement of more complex assemblies of mRNPs is implicated, including the manifestation of structures likely dependent upon liquid-liquid phase separation, described as membrane-less organelles, also called biomolecular condensates, such as stress granules[3]. These have been identified in diverse biological systems and act to sequester proteins and mRNAs, either restricting their translation or degradation or regulating their access to cytosolic signalling molecules[4–8]. Importantly, such biomolecular condensates are potentially dynamically rather than passively organised, and their assembly and disassembly can be influenced by distinct components, an example being the regulated dissolution of stress granules in mammalian cells by a protein kinase of the DYRK family[9–12]. In trypanosomes, stress granules have also been observed microscopically and seem to form particularly under conditions of nutrient starvation[13–19]. This has allowed the exploitation of methodologies whereby granule components can be identified after physical selection or affinity purification. One such example is through the characterisation of stress granules induced under glucose starvation in *T. brucei* where an innovative approach exploited the cytoskeletal corset of the parasite to act as a molecule sieve allowing granules to be purified and characterised by mass spectrometry[16].

Stress responses have been seen to be invoked in altered nutritional and thermal conditions in the laboratory. However, it has also been proposed that such stress responses represent an evolutionary path to the developmental progression of the parasite. Specifically, it has been suggested that the underlying stress responses generated an adaptation platform upon which appropriate developmental responses could be evolved as the parasites progressed from a monoxenic lifestyle to one involving mammalian hosts and insect vectors[20]. Consistent with this, several protein components have been visualised by microscopy to assemble into granular structures as trypanosomes undergo the transition between bloodstream and procyclic forms, the stage of the parasite that colonises the tsetse fly gut. Further, as trypanosomes prepare for transmission, they undergo a quorum-sensing-like response to accumulating environmental oligopeptide signals to generate so-called stumpy forms. These have exited the cell cycle, are morphologically distinct from proliferative bloodstream slender forms and are optimised for efficient differentiation to procyclic forms[21–26]. A molecular characterisation of the underlying machinery responsible for the quorum-sensing response identified a number of molecules that were predicted as gene regulators, and/or have been identified as components of the stress granules generated under glucose starvation[27,28]. Moreover, one signalling molecule identified as a regulator of quorum-sensing, TbDYRK, is a protein kinase that is phylogenetically related to the DYRK3 family of kinases involved in mammalian stress granule dissolution. These observations prompted us to explore the extent to which membraneless granule formation accompanied the differentiation of trypanosomes between slender and stumpy forms in their mammalian host, and the composition and positional dynamics of any component molecules in comparison to nutritional stress responses. Further, we investigated the extent to which granule formation was regulated in the presence and absence of the developmental regulator TbDYRK.

Here, we show that differentiation between slender and stumpy forms does involve the appearance of membraneless granules but that these differ in composition and positional dynamics from those generated by nutritional stress. Furthermore, our results highlight the role of TbDYRK in mediating the specificity and complexity of differentiation and stress granule assemblies.

## Results

### Bloodstream form trypanosomes generate cytoplasmic granules in response to glucose stress
To explore any contribution of TbDYRK to cytoplasmic granule formation we initially investigated the impact of glucose depletion on pleomorphic bloodstream form *T. brucei* AnTat1.1 in vitro. Thus, cultured parasites were incubated either in glucose-replete Creeks minimal medium with 10%FBS media or in media depleted of glucose as a supplement (albeit retaining glucose in the provided serum component, estimated to be 0.15 mM overall). The parasites were then fixed and analysed using an antibody detecting the Alba 3 protein which has been previously found to associate into cytoplasmic granules that colocalizes with poly(A+) RNA in procyclic form *T. brucei* under glucose starvation conditions[14–16]. Western blotting revealed that the antibody potentially detects both Alba 3 and Alba 4 (Supplementary Fig. 1) such that detected granules could comprise either or both proteins. Figure 1 demonstrates that a diffuse Alba 3/4 staining profile was observed with parasites growing in replete conditions, but that after 30 and 60 min of exposure to medium with reduced glucose, cytoplasmic granules were detected. Next, a similar analysis of the assembly of cytoplasmic granules was carried out in the absence of TbDYRK. To achieve this a previously generated TbDYRK knockout line was incubated either in the presence or absence of glucose for 30 and 60 min. As with wild type cells, this resulted in the appearance of granules detected by the Alba 3/4 antibody suggesting that TbDYRK was not a requirement for their formation (Fig. 1B). As further controls, an add-back line with TbDYRK expressed from its endogenous locus was generated, as was an equivalent line expressing a catalytically disabled TbDYRK S856G mutant. This mutation converts the atypical DFS motif in *T. brucei* TbDYRK to a conventional eukaryotic DFG motif that is normally associated with reorientation of the catalytic residues and ATP binding, resulting in an 87% reduction in kinase activity[28]. In both cases, Alba 3/4 was observed to coalesce into cytoplasmic granules, demonstrating that active TbDYRK is not necessary for Alba 3/4 +ve glucose depletion related granules to form.

### TbDYRK plays a role in stress granule dissolution during the recovery from glucose starvation
In other eukaryotes, DYRKs have been implicated in the dissolution of stress granules. Having demonstrated that TbDYRK was not required for granule formation we explored its contribution to stress granule dissolution after glucose re-addition to starved cells. To achieve this, we initially created a construct to express TbAlba 4 fused to Ty1 epitope tagged mNeonGreen allowing its unambiguous detection in case Alba 3 and Alba 4, which are both detected by the available Alba 3 antibody, behaved differently with respect to their granule assembly and dissolution. This was introduced into the wild type *T. brucei* AnTa1.1 J1339 cells or into the TbDYRK KO line such that the prevalence of Alba 4 containing granules could be monitored. Figure 2 shows the number and density of Alba 4 positive granules in parasites in glucose-replete conditions or after 35 min incubation in glucose depleted conditions, and thereafter with glucose re-supplemented for 10, 30 and 60 min. This examines the generation of cytoplasmic granules and their dissolution during recovery with the re-addition of glucose. As previously, in the presence of TbDYRK cytoplasmic granules appeared when depleted of glucose, with 79% of cells exhibiting at least 4 granules (median of number of granules in non-stress cells = 4.5). However, these granules rapidly reduced in number after re-addition of glucose, with only 71%, 19% and 34% of cells exhibiting at least one cytoplasmic granule after 10, 30 and 60 min respectively. In the TbDYRK KO line, granules were also detected in the presence or absence of glucose (33% and 56% of cells, respectively exhibited at least 4 granules). With the re-addition of glucose many cells retained cytoplasmic granules for at least 60 min and presented less variation than observed in the presence of TbDYRK, with 12%, 27% and 32% of parasites still possessing at least 4 granules after 10, 30 or 60 min, respectively. This indicated that granules could form upon glucose starvation in the absence of TbDYRK but their regulation and dynamism was affected, there being less variation observed upon stress, suggesting a possible role for the kinase in granule dissolution.

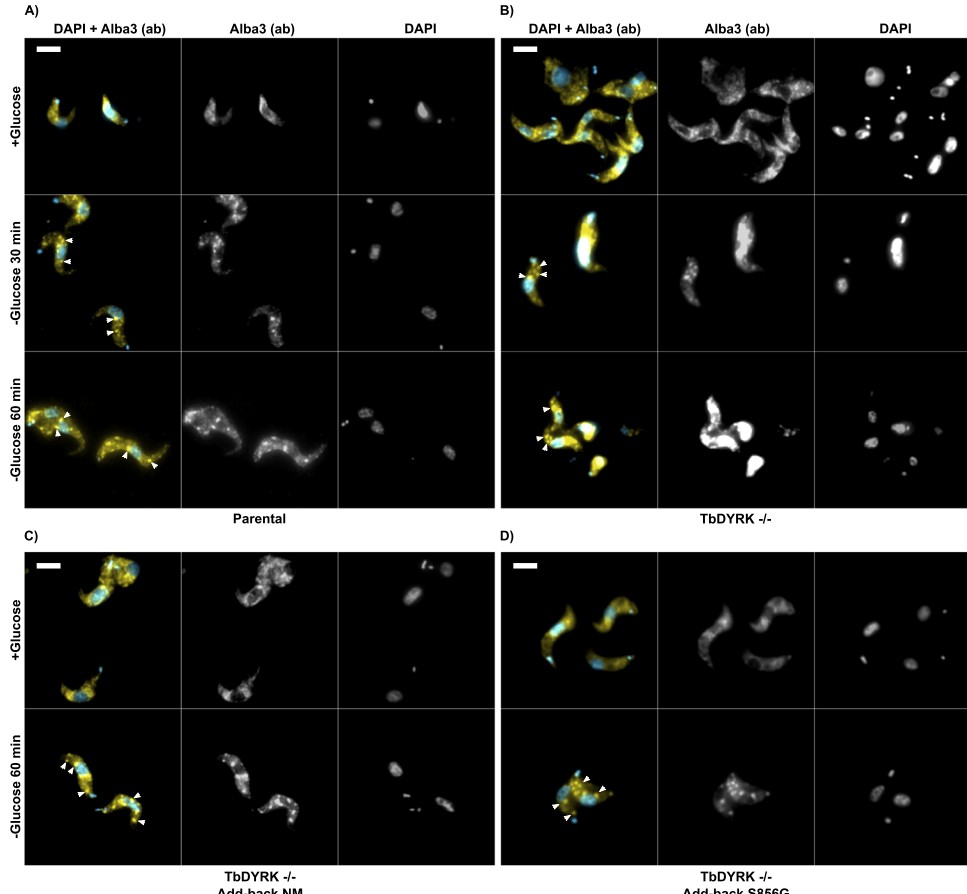

**Fig. 1 | Glucose starvation stress granules are present in the bloodstream form of *T. brucei* and their formation is independent of the presence and activity of TbDYRK in vitro.** Immunofluorescence of bloodstream slender cells exposed to glucose starvation for 30 min and 60 min in respective cell lines: (**A**) Parental = AnTat1.1 J1339, (**B**) TbDYRK-/- = AnTat1.1 J1339 TbDYRK-/-, (**C**) TbDYRK-/- Add-back NM = AnTat1.1 J1339 TbDYRK-/+(add-back; non mutated), (**D**) TbDYRK-/- Add-back S856G = AnTat1.1 J1339 TbDYRK-/+(add-back mutant S856G). Alba 3/4 localisation is revealed by anti-Alba 3 antibody (ab) (false coloured yellow), nucleus and kinetoplast using DAPI (false coloured blue). Scale bar = 10 μm. *n* = 3 independent experiments with similar outcome and no results were excluded.

## Cytoplasmic granules form during differentiation to stumpy forms

Although glucose depletion is regularly used experimentally to induce the appearance of stress related granules in *T. brucei*, we were interested whether granules were also characteristic of parasites undergoing physiological differentiation to stumpy forms, a transition also with the potential to invoke stress-related responses[20]. Therefore, we investigated the appearance and cellular distribution of granules in *T. brucei* AnTatT1.1 EATRO 1125 lines with Ty-mNeonGreen (TYmNG) fusions of either Alba 3 or Alba 4. These parasites were infected into mice, with bloodstream parasites isolated on days 4 to 6 of the infection as they underwent their quorum-sensing induced transition from slender to stumpy forms. At each time point, samples were isolated and monitored for the expression of the stumpy marker protein PAD1, using an antibody detecting that protein, or for the distribution of Alba 3 or Alba 4, using the BB2 antibody detecting the Ty1 epitope tag incorporated into each fusion protein. Figure 3 and Supplementary Fig. 2 demonstrate that as the cells progressed from being slender/intermediate (PAD1 negative) to being stumpy in morphology (PAD1 positive) the generation of cytoplasmic granules was observed. In PAD1 negative cells the distribution of either TYmNG::Alba 3 or Alba 4::TYmNG was diffuse. However, on D5 and D6 of the infection, as PAD1 positive cells appeared, discrete granules were formed in the cytoplasm between the cell nucleus and kinetoplast. Granules were then found in the perinuclear region and, thereafter clustered at the cell anterior side of the nucleus. This result was confirmed with the endogenous tagging of Alba 3 at either the N-terminal (Fig. 3) or in the C-terminal (Supplementary Fig. 2b) end of the protein.

To investigate whether the differentiation granules detected during stumpy formation were equivalent to granules induced during glucose depletion, we assayed the assembly of granules containing PABP1 and PABP2 which have been previously observed to assemble into granules upon glucose depletion[14,16]. Each protein was monitored, as with Alba 3 and Alba 4, by engineering cell lines expressing TYmNG tagged fusions of each. Thereafter, their distribution was assayed under conditions of glucose depletion (only PABP2 was tested in a glucose depletion assay) or in cells undergoing differentiation between slender and stumpy forms during the time course of an infection of the parasites in mice (both PABP1 and PABP2 tested). With glucose depletion, PABP2 redistributed from a diffuse cytoplasmic signal to discrete granules, similar to those observed with Alba 3 and Alba 4 under similar conditions (Supplementary Fig. 3a). However, PABP2 and Alba3/4 only partially colocalize in the absence of glucose with just 15 to 19.6% of the signal presenting colocalization using the Manders' correlation coefficient (Supplementary Fig. 3b). When parasites were analysed during their development to stumpy forms, a diffuse cytoplasmic signal was observed in both PAD1 negative slender cells and PAD1 positive stumpy cells and this profile was equivalent for both PABP2 and PABP1 (Supplementary Fig. 3c, d). This indicates that although PABP1/2 and Alba 3 and 4 assemble into granules upon glucose starvation that only partially colocalize, during differentiation their behaviour is different with only Alba 3 and 4 forming cytoplasmic

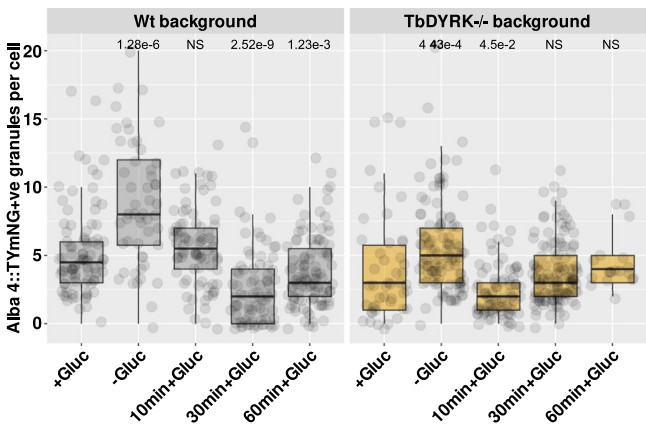

**Fig. 2 | TbDYRK controls glucose stress granule dynamism.** Quantification of Alba 4+ve granules in the Wt and TbDYRK-/- cell lines during the recovery from glucose starvation condition. The non-parametric Wilcoxon test was used to compare means of two independent samples. The null hypothesis states that the distributions of both populations are identical. Assuming that the responses are continuous, the alternative is restricted to a shift in location. *P* values are indicated when the null hypothesis is rejected; NS non-significant compared to the +glucose condition in the respective cell line. A detailed significance table and the number of values measured can be found in supplementary data 1. Bloodstream slender cells were exposed to glucose starvation for 35 min (-gluc) followed by addition of glucose for 10 (10 min + Gluc), 30 (30 min + Gluc) and 60 (60 min + Gluc). Cell line: Wt background = AnTat1.1 J1339 TbDYRK+/+ Alba 4::TYmNG, TbDYRK-/- background = AnTat1.1 J1339 TbDYRK-/- Alba 4::TYmNG. Alba 4 localisation in granules was revealed by anti-Ty antibody. The number of granules per cell was automatically evaluated using a script run in Fiji using the following parameters for the size = 0.12–1 μm and the circularity = 0.7–1. Boxplots represent the interquartile range (IQR) from the 1st (25th percentile, Q1) to the 3rd (75th percentile, Q3) quartile, the median and whiskers indicate the maximum (Q3 + 1.5*IQR) and minimum (Q1 − 1.5*IQR) values. Individual data points are shown using overlaid dot plots. Each dot represents one parasite. *n* = 3 biological replicates per *n* = 2 independent experiments were performed, no results were excluded.

granules. This indicated that the composition and assembly of granules differed in cells exposed to reduced glucose and those undergoing the differentiation to stumpy forms.

### Differentiation granules dynamically appear and localise during development to stumpy forms

To explore the composition and assembly of cytoplasmic granules during the differentiation from slender to stumpy forms we generated a panel of cell lines expressing a TYmNG fusion of different potential stress granule components. This included the aforementioned PABP1 and 2 (Supplementary Fig. 3), Alba 2 (Supplementary Fig. 4), 3 and 4 (Fig. 3 and Supplementary Fig. 2) and also a previously characterised granule component XRNA (Supplementary Fig. 5), a substrate of TbDYRK of unknown localisation, ZC3H20[28], and a cytoplasmic control NEK17[29] (Supplementary Fig. 6). These 8 tagged cell lines were then individually infected into mice and their assembly into cytoplasmic granules and cellular location monitored by microscopy. Specifically, cells were monitored for their expression of PAD1 (using an antibody detecting the PAD1 protein) and for the assembly and cellular location, of each reporter protein into cytoplasmic granules, this being further assessed with respect to the kinetoplast to nuclear distance (which declines as cells transition from slender to stumpy forms). Each cell was scored for the number of granules detected, the intensity of the granules and their positioning either anterior or posterior to the cell nucleus, in a perinuclear position or for their appearance at a proximal location of the STuRN (stumpy regulatory nexus), a posterior site where several regulatory molecules have been found to gather in stumpy forms[30]. These results for the respective markers are

summarised in Fig. 4A, B. Figure 4A and Supplementary Figs. 4–7 demonstrate that Alba 3 was observed to form granules at day 4 post infection (PI), followed by Alba 4 at day 5 (Supplementary Fig. 2), these representing the most numerous and intense granules of the suite of markers analysed. Alba 2 also formed granules, although this was weaker and observed only from day 6 PI (Supplementary Fig. 4); XRNA granules also appeared from day 6 when the population was predominantly stumpy in morphology (Supplementary Fig. 5). Of the remaining markers, ZC3H20 (not shown), NEK17 (Supplementary Fig. 6) as well as PABP1 and 2, as observed in Supplementary Fig. 3, did not assemble into detectable granules. Using the mean of fluorescence per cell as a proxy for protein expression we observed little variation over time for proteins such as Alba 2, Alba 4 and XRNA suggesting that these proteins accumulate to differentiation granules with no change of their abundance during differentiation (Supplementary Fig. 7). This is different from PABP1 that accumulates in the cytoplasm while the cells are expressing PAD1 and from Alba 3 that both accumulates in granules and presents an increase of abundance during the time course of the differentiation.

Analysing the distribution of the granules generated during differentiation revealed further evidence for their dynamic positioning during the developmental process, this being additionally correlated with the K-N dimension in each cell (Fig. 4B). Early in differentiation (day 5 PI; yellow symbols in Fig. 4B) Alba 3 and Alba 4 granules were detected between the nucleus and cell posterior or in a perinuclear position. In contrast, by day 6 (blue symbols) of the infection, granules were also detected between the cell nucleus and cell anterior. The posterior location of granules in early time points did not correspond to the STuRN since co-labelling for the STuRN marker protein TbPIP39[30], a serine threonine phosphatase, did not reveal co-distribution with the granules detected for the tagged Alba 3 (Supplementary Fig. 8). XRNA was positioned either at the cell posterior as previously described[19] or in a region close to the cell posterior near the STuRN at later differentiation stages but did not colocalize with TbPIP39 (Fig. 4B, Supplementary Fig. 5c). Notably the redistribution and assembly of the marker proteins Alba 3, Alba 4 and XRNA into granules was overwhelmingly seen in cells that expressed the PAD1 stumpy marker protein (circle symbols on Fig. 4B) and few were seen in cells that remained PAD1-ve (triangle symbols on Fig. 4B), demonstrating that the granule assembly was associated with the differentiation process.

### Disruption of differentiation with TbDYRK knockout also perturbs differentiation granules

TbDYRK depletion or knockout was previously reported disrupting slender to stumpy differentiation[27,28]. To investigate the link between differentiation and granule dynamism, we endogenously labelled Alba 4 with a Flag-APEX2 tag in both a WT (TbDYRK+/+) and a TbDYRK knockout (TbDYRK-/-) background and infected mice with the resulting cell lines. We followed by immunofluorescence the parasite differentiation during the time course of the infection using an anti-PAD1 antibody, and the formation of Alba 4+ve differentiation granules using an anti-Flag antibody. As previously described, TbDYRK-/- cells were unable to differentiate as revealed by the absence of PAD1 staining (Fig. 5A). We also observed a delay in the generation of Alba 4+ve granules that only appeared at a late time point, at day 6 PI, and these were located throughout the cytoplasm (Fig. 5A). Automated counting of Alba 4+ve granules confirmed that they rapidly and strongly accumulate in TbDYRK-/- in PAD1 negative cells at day 6 PI, while their numbers increase more modestly at Day 5 and remain stable in the TbDYRK+/+ cell line while cells are differentiating and express PAD1 (Fig. 5B). Notably, we observed that cells that did not differentiate at day 6 PI in the TbDYRK+/+ background exhibited a reduction of Alba 4+ve differentiation granules compared to day 5 (compare the PAD+ve and PAD-ve cells in Fig. 5B). Altogether, these

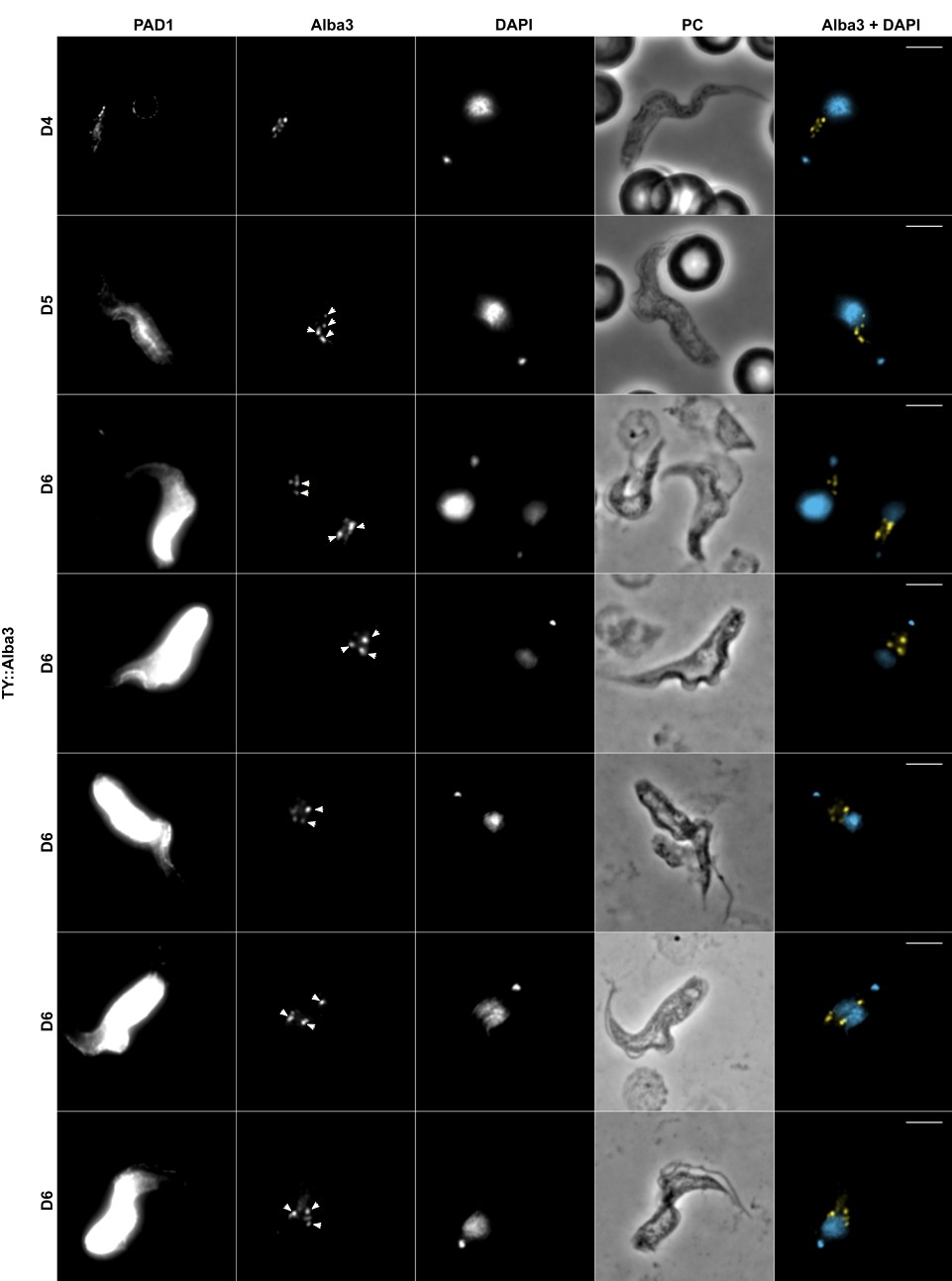

**Fig. 3 | Granules are formed and change localisation during quorum-sensing differentiation in vivo.** Immunofluorescence of blood-smears of mice infected with a TYmNG::Alba 3 tagged cell line from day 4 (D4) to day 6 (D6) post-infection. $n = 3$ animals were infected per $n = 2$ independent experiments, no results were excluded. PAD1 expression is revealed using anti-PAD1 antibody (PAD1), Alba 3 localisation is revealed by anti-Ty antibody (Alba 3, false coloured Yellow), nucleus and kinetoplast using DAPI (false coloured Blue). PC Phase contrast. Acquisition settings were kept constant to detect weak positive cells, leading to saturation of the PAD1 signal on late time points. Scale bar = 10 μm. Arrows highlight differentiation granules.

results suggest that the timely generation and cytological distribution of differentiation granules accompanies the development into stumpy cells and that TbDYRK may contribute to this.

## The composition of glucose and differentiation granules in the presence or absence of TbDYRK

Having determined that glucose starvation granules and differentiation granules were different and that their timely generation and composition during development could be influenced by TbDYRK we aimed to identify the molecular composition of both starvation and differentiation granules. This was achieved using an APEX2 proximity labelling approach[31], which has been successfully used to identify components of membraneless organelles in human neurons[32]. APEX2

is an engineered monomeric ascorbate peroxidase that converts biotin-phenol into a short-lived biotin radical that interacts with nearby proteins, resulting in the rapid (within 1 min) covalent attachment of a biotin tag in the presence of hydrogen peroxidase. Initially, we confirmed the most suitable conditions of biotin-tyramide and hydrogen peroxidase treatment in trypanosomes using the flagellar protein DRC1 as a test (Supplementary Fig. 9a), since this was previously used to characterise flagellar tip composition in *T. brucei*[33].

Once optimised, the biotinylation enzyme, coupled to 2 flag tags, was fused to either Alba 4 or Alba 3 as bait proteins. Thereafter, the resulting biotinylation patterns were identified by western blot in a WT or TbDYRK knockout background in the presence or absence of glucose, to induce starvation conditions (Supplementary Fig. 9b, c).

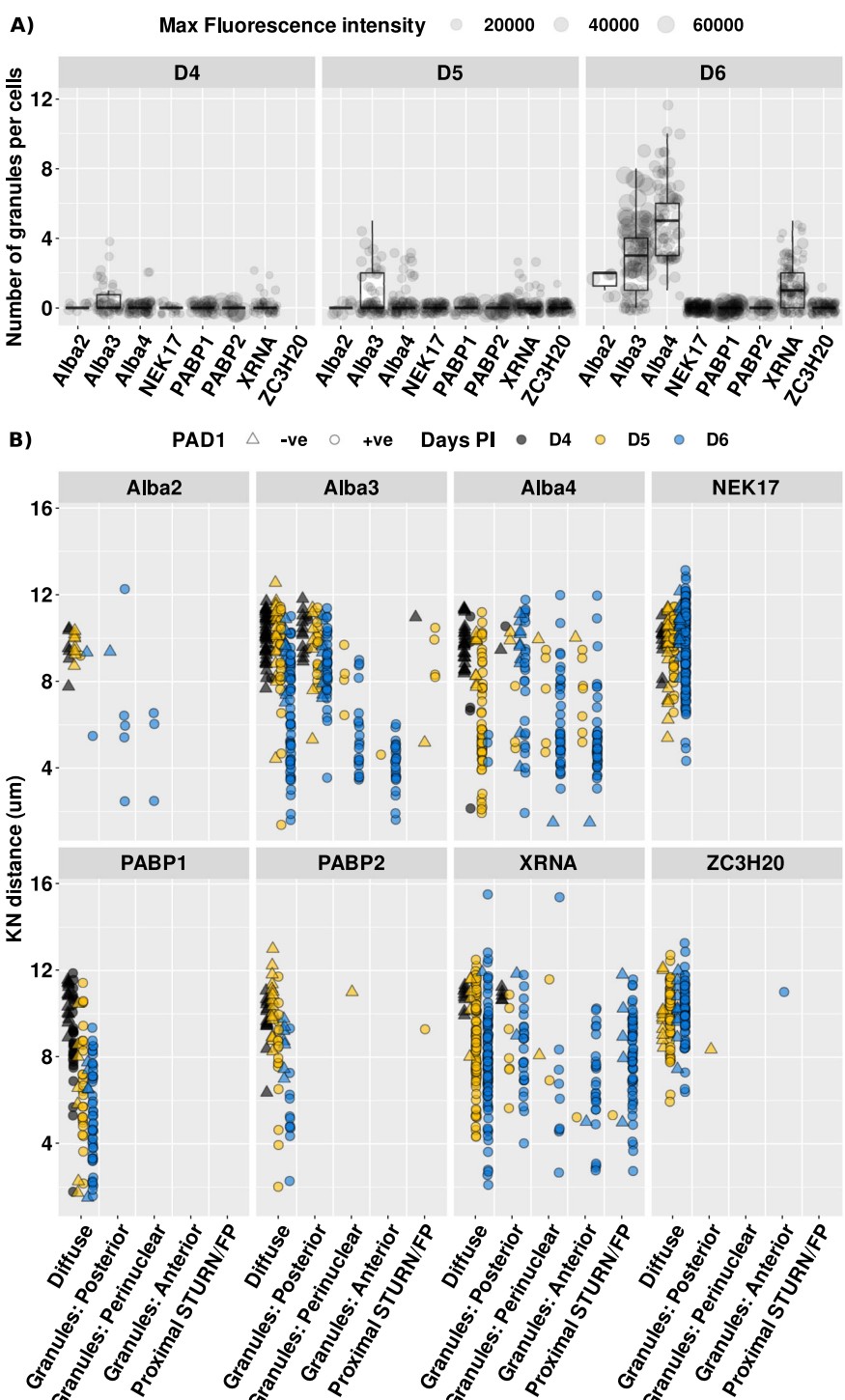

**Fig. 4 | Localisation and composition of differentiation granules is dynamic during quorum-sensing differentiation. A** Quantification of Ty1-positive granules in the different cell lines during the time course of an infection at days D4, D5, D6 post-infection. $n = 3$ animals were infected with each cell lines over $n = 3$ independent experiments, no results were excluded. Boxplots represent the interquartile range (IQR) from the 1st (25th percentile, Q1) to the 3rd (75th percentile, Q3) quartile, the median and whiskers indicate the maximum (Q3 + 1.5*IQR) and minimum (Q1 − 1.5*IQR) values. Individual data points are shown using overlaid dot plots. Size of the dots represent the maximum of intensity measured in the corresponding cell. **B** Representation of Ty tag protein localisation. Diffuse - in granules either posterior to the nucleus, perinuclear or anterior to the nucleus−or in proximity to the STuRN/Flagellar pocket (FP), in relation to the distance between the kinetoplast and nucleus (KN distance in μm) in 1K1N cells−used to determine the relative stage of stumpy differentiation. Colours represent the time points in days post-infection (Days PI) and shapes represent the PAD1+ staining, being either negative (−ve, triangle) or positive (+ve, circle).

Notably, Alba 4::APEX2 itself was not identified in the biotinylated fraction by western blotting after streptavidin magnetic bead enrichment, suggesting the absence of accessible lysines for biotinylation on this protein.

Biotinylated enriched fractions of Alba 4::APEX2 in WT or TbDYRK-/- cell lines, exposed or not to 35 min glucose starvation, were then analysed by mass spectrometry and compared to the results obtained from the parental cell line treated under the same conditions.

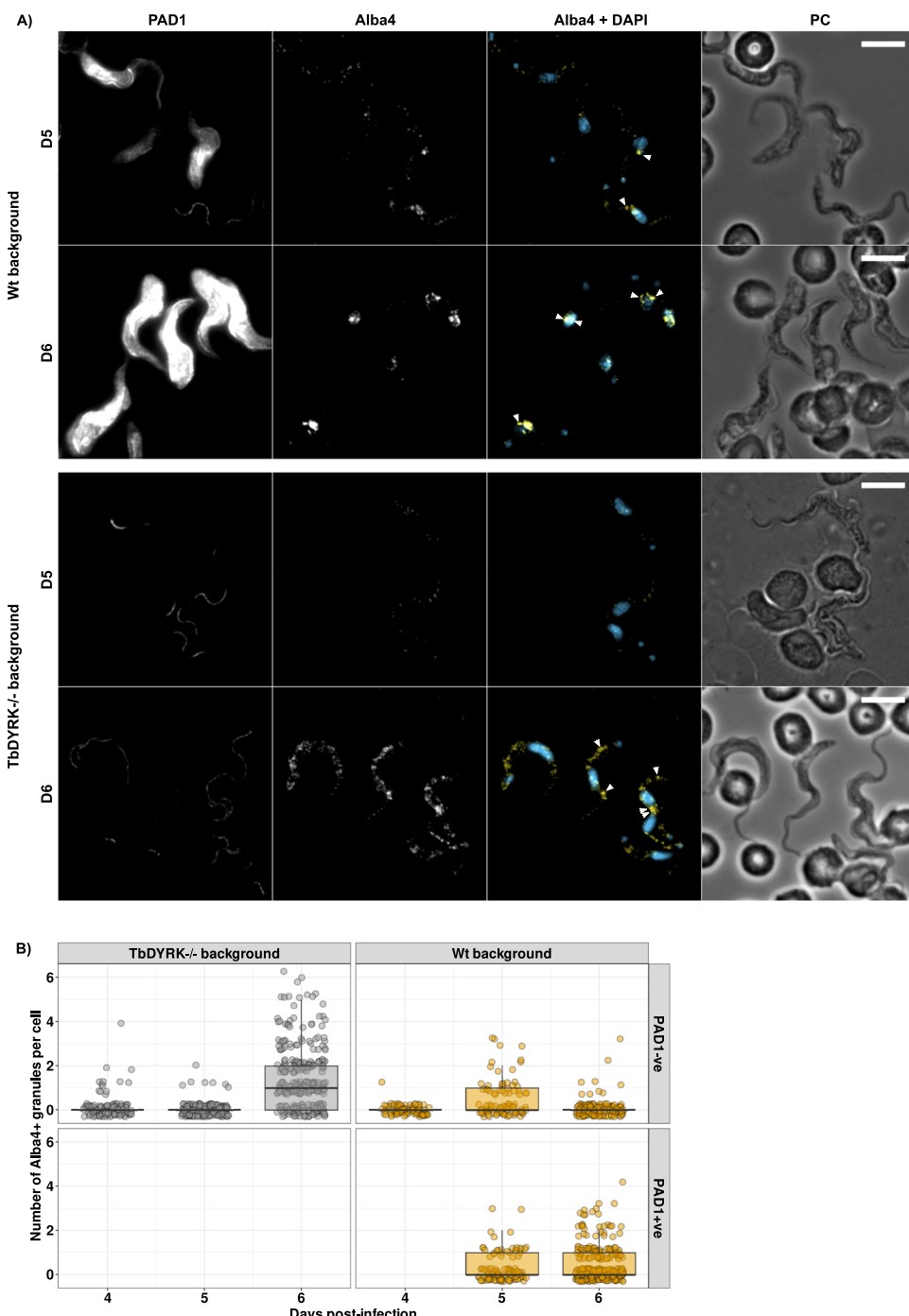

**Fig. 5 | Deletion of TbDYRK leads to uncontrolled accumulation of differentiation granules during quorum-sensing differentiation.**
**A** Immunofluorescence of blood-smears of mice infected with Alba 4::APEX2-Flag tagged cell line, in TbDYRK+/+ (Wt) or TbDYRK-/- genetic background, at day 5 (D5) and day 6 (D6) post-infection. $n = 3$ animals were infected with each cell lines in $n = 1$ experiment. PAD1 expression is revealed using anti-PAD1 antibody (PAD1), Alba 4 localisation is revealed by anti-Flag antibody (Alba 4, false coloured Yellow), nucleus and kinetoplast using DAPI (false coloured Blue). PC Phase contrast. Acquisition settings were kept constant to detect weak positive cells, leading to saturation of the PAD1 signal on late time points. Scale bar = 10 µm. Arrows highlight differentiation granules. **B** Quantification of Flag positive granules in the different cell lines during the time course of the infection relative to the expression of the stumpy differentiation marker PAD1 negative (PAD1-ve) or positive (PAD1+ve). Boxplots represent the interquartile range (IQR) from the 1st (25th percentile, Q1) to the 3rd (75th percentile, Q3) quartile, the median and whiskers indicate the maximum (Q3 + 1.5*IQR) and minimum (Q1 − 1.5*IQR) values. Individual data points are shown using overlaid dot plots.

For each condition, three independent replicates were analysed and those that showed statistical significance identified. Note that 1 replicate for each of the following conditions has been removed from statistical analysis due to poor data quality: Alba 4 in TbDYRK-/- background under both 'no stress' and starvation conditions, parental cell line under starvation. Principal component analysis (PCA) of the remaining samples (Supplementary Fig. 10a) indicates a good reproducibility between replicates with the greatest variation observed in the asynchronous +glucose populations and the unspecific control J1339 parental cell line. Under 'no stress' conditions, we identified 15 proteins proximal to Alba 4 in the WT background (Fig. 6A, yellow circles). These include proteins involved in signal transduction and

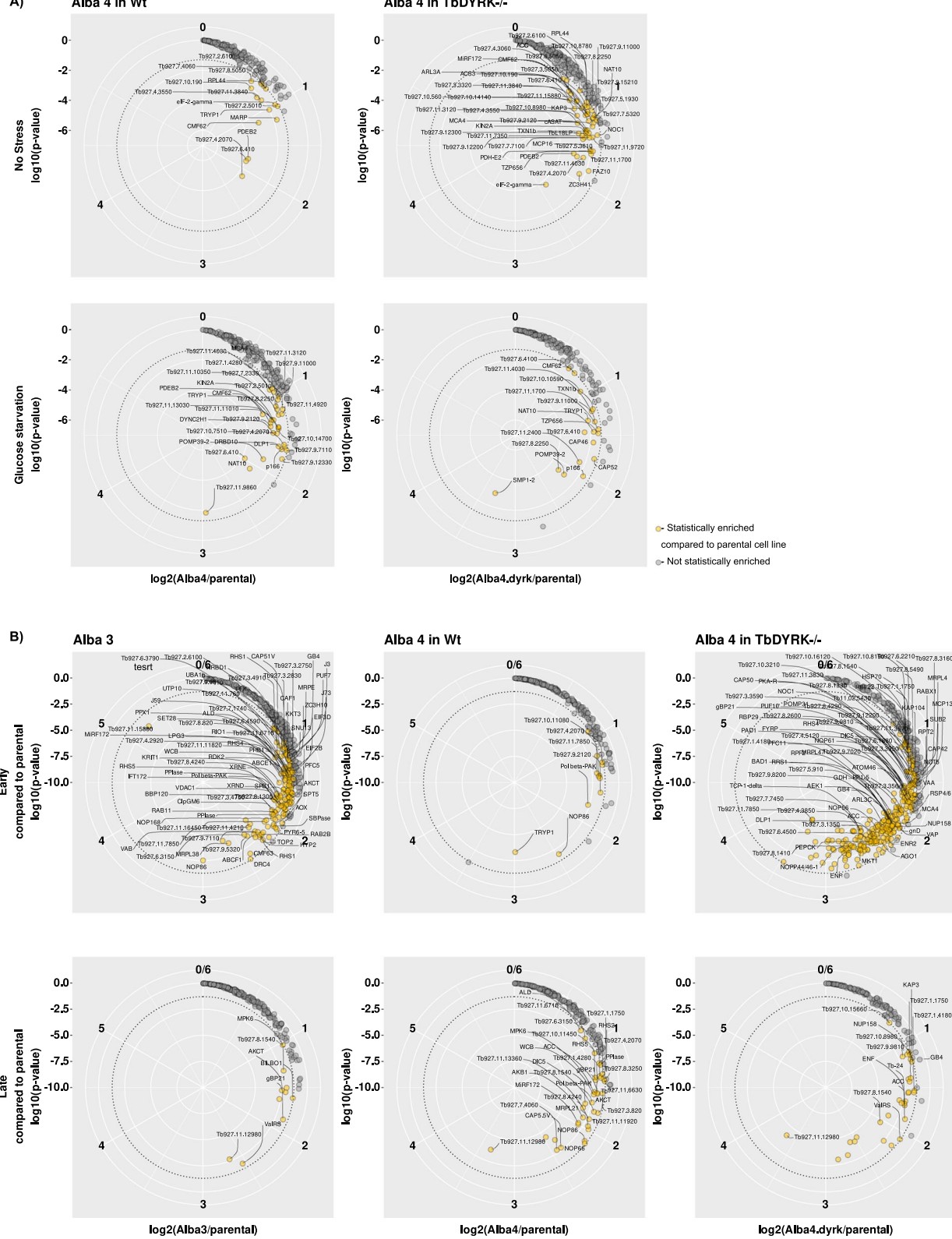

**Fig. 6 | Dynamic compositions of starvation stress and differentiation granules are regulated by TbDYRK.** Circular plots representing proteins proximal to Alba 3 and Alba 4 in Wt background or Alba 4 in TbDYRK background identified by mass spectrometry. Circular axes represent the log2 fold change enrichment in the bait protein compared to the parental cell line. The y-axis is the -log10 of the $p$ value. Statistical enrichment analysis was performed using two-sided moderated t-statistic as described in the Methods section. Proteins presenting a fold change >1.5 and a $p < 0.05$ in a given condition were considered statistically enriched.

Statistically enriched proteins proximal to the bait protein are highlighted in yellow. **A** Stress granule composition in vitro in non-stress (top panels) or glucose starvation (bottom panels). Numbers of independent biological replicates used for analysis: Parental no stress $n = 3$, parental starvation $n = 2$, Alba 4 in Wt no stress $n = 3$ and starvation $n = 3$, Alba 4 in TbDYRK-/- no stress $n = 2$ and starvation $n = 2$. **B** Differentiation granule composition at two different time points during the course of mice infection, early (top panels) and late (bottom panels).

metabolism control such as PDEB2, Tb927.2.5010, Tb927.8.5050 (an OTU-like cysteine protease), Tb927.7.4060 (a further cysteine protease) and TRYP1, 4 proteins involved in translation control (eIF-2-gamma, RPL44, Tb927.10.190, Tb927.4.3550) and 2 microtubule/flagella associated proteins (CMF62 and MARP). With glucose starvation, we observed a drastic change of the proximal proteome of Alba 4 confirming the change of localisation of many proteins to stress granules, although this was not reflected in the enrichment of any particular GO term. Overall, 30 proteins were identified in this condition with only 6 in common with the non-stressed condition (PDEB2, Tb927.2.5010, TRYP1, CMF62, Tb927.6.410, Tb927.4.2070) indicating that these proteins may be ubiquitous interactors with Alba 4 in culture conditions. Other Alba 4 stress granule proximal molecules included cytoskeleton/flagella interacting proteins (DLP1, KIN2A, DYNC2H1, Tb927.11.9860, MCA4−a catalytically inactive metacaspase[34], calmodulin), mitochondria-associated proteins (p166, POMP39-2) and nucleotide-binding proteins (NOG1, DRBD10, NAT10, Tb927.8.2250−a tRNA ligase phosphodiesterase, Tb927.10.7510−a chromatin-binding protein) (Supplementary data 2). In the absence of DYRK a greater number of Alba 4 proximal proteins were detected in both non-stressed and starved cells, with 51 and 18 proteins identified respectively (Fig. 6A, right). As in the WT background cell line, no GO term enrichment was detected under starvation conditions, while an enrichment for 'structural constituent of ribosome' was observed in the non-stressed condition.

We next compared the composition of glucose starvation stress granules with the differentiation granules formed during development in vivo. To do so, we purified parasites expressing the bait proteins Alba 3::APEX2 or Alba 4::APEX2 in a WT or TbDYRK null background, from infected mice at 2 time points: (1) an 'early' time point representing a slender replicative population with a parasite density <1e8 parasites/mL and 0 % PAD1 expression, and (2) a 'late' time point representing a differentiating (WT) or high parasitaemia (TbDYRK-/-) population with a parasite density >2e8/mL and, for the WT cells, >40% PAD1+ve cells. TbDYRK-/- cells are unable to differentiate and so do not express PAD1 (Fig. 5)[28]. Immediately following their purification from blood, parasites were exposed to biotin-tyramide in complete CMM media[35] at 37 °C for 30 min and then exposed to 1 min treatment with hydrogen peroxide before streptavidin enrichment, trypsin and lys-c digestion and mass spectrometry analysis. As previously, the parental cell line (with no tag on either bait protein) treated in the same conditions was used as a specificity control. Three independent replicates were used and analysed for each condition. The PCA presented in Supplementary Fig. 10b reveals good reproducibility between replicates, with the early time point showing the greatest variation, likely due to the lack of population synchrony and the TbDYRK-/- cell line clustering separately at both time points.

A brief overview of the data confirmed our previous observations (shown in Fig. 4A) that Alba 3+ve granules appear first during differentiation, with 86 Alba 3 proximal proteins compared to 12 proteins for Alba 4 at the early time point (Fig. 6B). We then identified 10 and 35 proteins, respectively, proximal to Alba 3 and 4 at the late time point. In the absence of TbDYRK, more Alba 4 proximal proteins were detected in both early and late stages, with 287 and 31 proteins identified respectively (Figs. 6, 7). GO term analysis revealed that Alba 3+ve granules in parental cells were enriched for pyrimidine nucleobase biosynthetic and fructose 6-phosphate metabolic processes while in the absence of TbDYRK, both Alba 3+ve and Alba 4+ve granules were enriched for pyrimidine nucleobase biosynthetic processes, as well as translational regulation processes (maturation of LSU-rRNA and ribosomal large subunits). To further explore these observations, we compared the proximity proteomes under starvation and differentiation conditions and identified little overlap between Alba 4 starvation stress granules and Alba 4 differentiation granules (4/12 proteins, Supplementary Fig. 11a). However, 14 proteins that were found in both

the Alba 3 proximity dataset in a parental background and the Alba 4 proximity dataset in a TbDYRK-/- background were also found in starvation stress granules proximal to Alba 4. These observations suggest that early Alba 3+ve granules and Alba 4+ve granules in the absence of TbDYRK are enriched for stress response proteins.

In the parental background, both Alba 3+ve and Alba 4+ve granules presented 10 and 8 common proteins, respectively in early and late time points (Fig. 7 and Supplementary data 2). Additionally, proteins such as the glycine acetyl transferase AKCT, the MAP kinase MPK6, the minicircle replication factor 172 MiRF172, the retrotransposon hot spot protein 5 (RHS5), the cyclophilin-type peptidyl-prolyl cis-trans isomerase (PPIase), the cytoskeleton associated protein WCB, the 3-methylcrotonoyl-CoA carboxylase beta subunit (Tb927.11.6630), the AAA ATPase (Tb927.11.13360) and the splicing factor 3a (Tb927.6.3150) were first found associated to Alba 3+ve granules at the early time point and then associate to Alba 4+ve differentiation granules in the later time point. We then grouped the proximal proteins according to their localisation in bloodstream forms as identified by ref. 36. (Supplementary Fig. 13). During the early time point, proteins from all compartments are targeted to Alba 3+ve granules. At a later time point we observed a relatively low number of proteins from the flagellum, proteasome and nucleus. Common proteins identified proximal to both Alba 3 and 4 were localised in slender forms in the endoplasmic reticulum, associated to microtubule structures or from the mitochondria and secretory/endocytic system. Together, these observations indicate that Alba 3+ve granules are formed early during the time course of an infection in response to the environmental host change, that the range of proteins targeted to the granules is initially broad in terms of protein localisation and that these granules may then be used as a precursor for the formation of Alba 4+ve granules, that appear in the later stage with cells differentiating into the stumpy form.

## Substrates of TbDYRK are associated with differentiation granules

Finally, we questioned whether substates of TbDYRK are targeted to differentiation granules during the time course of an infection by comparing our APEX proximity dataset to the list of TbDYRK potential substrates identified via phosphoproteome enrichment[28]. Three proteins were identified at the late time point that are potential substrates of TbDYRK; Tb927.9.1750 (in proximity to Alba 3 and 4 in parental and TbDYRK-/- cells), WCB and Tb927.1.4280 (in proximity to Alba 4 in parental and TbDYRK-/- cells). At the early differentiation time point, potential substrates such as SPB1, CAP51V, Tb927.7.3740 were identified proximal to Alba 3 in WT cells and to Alba 4 in TbDYRK-/- cells and LRRP1, Tb927.11.3490 and NOT5 were specifically targeted to Alba 4+ve granules in the absence of TbDYRK (Supplementary Fig. 12, Fig. 7 and Supplementary data 2). It is interesting to note that 5 of these proteins have low complexity regions[37], consistent with their association with membraneless granules.

## Discussion

The response of trypanosomes to developmental cues culminating in the generation of stumpy forms is often considered to have similarities to the response of the parasites to stress, including glucose starvation. In the work described here, we have analysed the assembly of several components identified as associated with stress granules in these parasites under conditions of either nutritional restriction (glucose starvation) or during their progression to stumpy forms in vivo. Further, we have explored the assembly of factors into cytoplasmic granules in the presence or absence of TbDYRK, a protein kinase implicated in the response to quorum-sensing and related to molecules in other eukaryotes that function in the dissolution of stress granules. This has demonstrated that the components associating with granules differ under glucose stress or differentiation conditions and

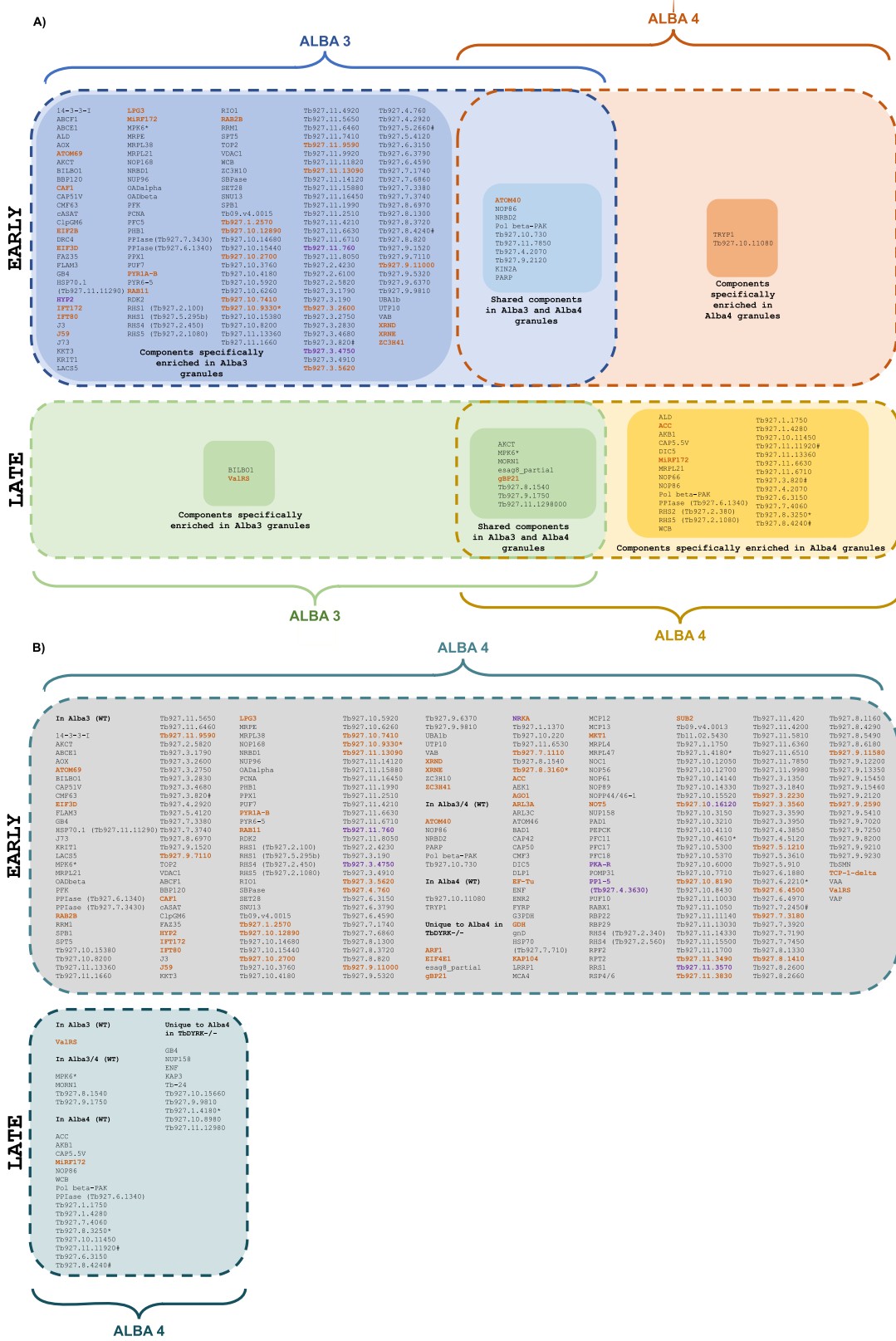

**Fig. 7 | Summary of the proteomes of differentiation granules proximal to Alba 3 or Alba 4.** Proximal proteomes of Alba 3 and 4 were analysed in a WT (**A**) or of Alba 4 in a TbDYRK-/- cell background (**B**). Proteins previously identified as part of stress granules are in Orange[16], potential interactors with Alba 3 with # (procyclic) and * (stumpy)[38], known and suspected components of the quorum-sensing pathway in Purple[27,29,66–70]. When available, common names are indicated in the figure for readability. Corresponding accession numbers can be found in Supplementary File 2.

furthermore revealed a progressive hierarchy in the composition and cytological location of granules as parasites transition from slender to stumpy forms in vivo. Also, it has revealed that in the absence of TbDYRK the disassembly of granules under glucose starvation is affected and that the composition of granules becomes more promiscuous. We anticipate that the role of TbDYRK in the recruitment or retention of granule components during differentiation may contribute to the loss of the ability to generate stumpy forms in its absence[27,28].

By following the progressive assembly of 8 potential stress granule components during differentiation, we were able to map the recruitment of components to distinct granules and also track their cellular location. This revealed that Alba 3 and Alba 4 were recruited to granular assemblies during differentiation, with Alba 3 slightly preceding Alba 4 in the developmental time course. This confirms previous observations that Alba 3 could be a core component of alba protein complexes[14]. Late in the process, XRNA also showed evidence of involvement in granules at the posterior location previously mapped for this component in procyclic forms[19]. XRNA was not identified proximal to Alba 3 or Alba 4 positive granules during differentiation whereas other 5′−3′ exoribonucleases XRND and XRNE were found proximal to alba 3 at the early time point during differentiation, suggesting their presence in different types of granules. We observed that in the early time point of differentiation proteins are targeted to granules from everywhere in the cell and that in the later time point this range is more restricted. Other components that are characteristic of glucose starvation granules identified in the procyclic stage of the parasites, such as PABP1 and 2, did not coalesce during differentiation, highlighting that different granules have different compositions dependent on the environmental cue. APEX2 driven biotinylation when fused to either Alba 3 and Alba 4 revealed relatively little overlap in the proximity proteomes of the two proteins, with only 9 components shared prior to the differentiation to stumpy forms and 8 components late in differentiation, with no overlap between the stages of development. This is consistent with the independent roles of Alba 3 and 4 in parasite development identified by gene knockout/RNAi and overexpression studies[15,38], although there is redundancy between the proteins in the bloodstream since Alba 4 alone can compensate for the combined absence of Alba 3 and 4. Nonetheless, both showed a similar positional relocation of their granules during differentiation using 2 different tags and at both N-terminal and C-terminal ends. Such granules relocate from a perinuclear position, as differentiation progresses (day 5), to a site between the cell nucleus and cell anterior. This did not match the distribution of Alba 3 and Alba 4 reported by ref. 38, where a location reminiscent of the STuRN (stumpy regulatory nexus[30];) was detected in stumpy forms. However, these were generated at high density with 1.1% methyl cellulose, as opposed to the physiological in vivo development we characterised here, suggesting a different compositional nature of mRNA/protein granules depending on the environmental cues. The respective tags incorporated into the respective proteins may also contribute to positional differences observed (N-terminal HA tag in Bevka et al.; C-terminal Ty/APEX2 tags and both N- and C-terminal Ty tag here).

Proximity profiling of the granules observed early and late in differentiation suggested a changed emphasis, with more proteins detected for Alba 3 early on and more proteins proximal to Alba 4 detected later in development. Notably, in this analysis, neither Alba 3 nor Alba 4 proteins were detected, likely because these are small proteins (20.8kDA and 22.7 kDa respectively) potentially less susceptible to biotinylation, especially if the C-terminal APEX2-flag tag interferes with any lysines accessible for biotinylation. Moreover, both proteins present a large number of possible trypsin and Lys-C cleavage sites (19.4% and 17.1% of the proteins respectively are either lysine or arginine) this being 166% and 146% higher than the overall total proteome of *T. brucei* (11.7% of lysine and arginine). This may lead to a reduction in the number of Alba 3 and 4 derived peptides identifiable by mass spectrometry. Of those proteins labelled through their proximity to Alba 3 and 4, GO term analysis did not identify an enrichment of particular functional groups and their involvement in the differentiation process awaits individual analysis.

In mammalian cells, DYRK3 is involved in stress sensing and regulates mTORC1 signalling through phosphorylation of the mTORC1 inhibitor PRAS40, which prevents its binding to mTORC1. DYRK3 also controls stress granule disassembly, exposing mTORC to activators such as lysosomal amino acids. Other DYRK kinases are also linked to both cellular stress responses and mRNP granules[39] as well as stress granule dissolution[40]. In our experiments, with the deletion of TbDYRK, granules continued to form under both glucose starvation and differentiation conditions, demonstrating that this protein is not required for granule formation. However, the intensity of granules generated under each condition was elevated and their dissolution upon the restoration of glucose was reduced—supporting a role for the protein in granule dynamics. This was matched by the complexity of the proximity proteome observed in the TbDYRK null background when Alba 4 was fused to APEX2. Here, the number of proteins found to be labelled early in development expanded from 11 proteins to ~300. About half of these proteins were shared with the protein set identified as proximal to Alba 3 in wild type parasites, whereas the rest were exclusively in proximity to Alba 4. Apparently, therefore, the specificity of the Alba 4 granule composition was reduced in the absence of TbDYRK (Supplementary Figs. 11 and 12) and this is consistent with the identification in these granules of many stress granule components previously identified by Fritz et al. in glucose starved procyclic cells[16]. When proteins targeted to Alba 4+ve granules in TbDYRK null cells were specifically examined early in differentiation, we noted the presence of some molecules known to be regulators of trypanosome quorum-sensing or differentiation, and also substrates of TbDYRK kinase identified by previous phosphoproteomic analysis. This included Not5, which was previously described as a potential 'slender retainer', i.e., a molecule that prevents the development from slender to stumpy forms. In one scenario, its inclusion in the Alba 4+ve granules in the absence of TbDYRK might protect Not5 from degradation such that it can contribute to the inhibition of differentiation. Alternatively the absence of TbDYRK may prevent the inhibition of Not5 by phosphorylation, preventing stumpy formation.

Our analysis of the formation and compositional changes of mRNA/granules during the quorum-sensing-dependent differentiation of *T. brucei* enabled us to propose the following working model (Fig. 8). Stress granules observed with glucose starvation are composed of molecules such as Alba proteins and PABPs. The nature of the

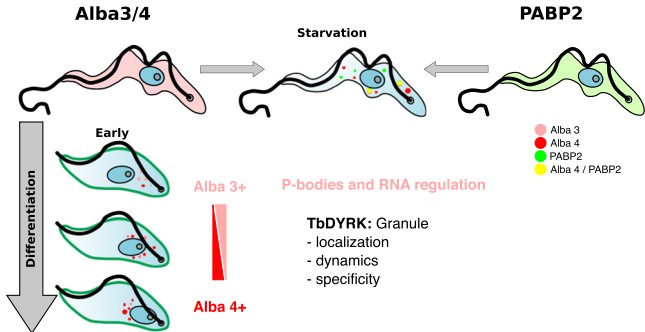

**Fig. 8 | Model for the assembly of granules during differentiation.** In stress conditions, Alba 3/4 and PABP2 proteins assemble into cytoplasmic granules that partially colocalize. In differentiation, in contrast, PABP2 is not incorporated into granules, with Alba 3 and then Alba 4 predominating as the parasites transition from slender to stumpy forms. TbDYRK is not needed for granule formation but regulates granule localisation, assembly dynamics and specificity.

starvation stress, such as glucose starvation of bloodstream forms, as in our study, or complete starvation of procyclics forms in PBS, as previously published, is likely to generate compositional changes of the stress granules. This is exemplified by the absence of DHH1 in the proximal Alba 4 starvation proteome in our analysis, while previously reported to colocalize with Alba 4[15]. Membraneless granules formed during *T. brucei* differentiation exhibit temporally regulated compositional changes with Alba 3 being first targeted to granules and possibly acting as core structure for the formation of later granules. Such early differentiation granules may serve as a fast 'stress' response to environmental changes and regulate mRNA expression, with their formation being independent of the regulation by TbDYRK phosphorylation. These granules may then evolve toward Alba 4+ve granules and potentially be involved in diverse regulatory processes with TbDYRK involved in their specificity and distribution. Later in the development of stumpy forms, Alba 2 and XRNA accumulate into granules and other membraneless organelles. Therefore, a possible role of differentiation granules could be to 'capture' proteins such as Alba 3 and 4 to enable the timely control of their association with polysomes, as has been observed during stumpy to procyclic differentiation. Interestingly, Subota et al., have observed the formation of Alba 3+ve differentiation granules during the differentiation from mesocyclic trypomoastigotes to epimastigotes[15], suggesting that the mechanisms observed in our study may be generalised for all differentiation stages. Furthermore, it has been proposed that Alba 3 may assist escape from translational repression during differentiation[38] and such function could happen through the biomolecular condensate we are observing.

Several types of biomolecular condensates have been associated with differentiation and developmental processes. For example, germ granules undergo regulated changes in composition, morphology, and localisation during *C. elegans* development. Specifically, P granules are thought to play specific roles with cytoplasmic P granules in early embryos that may assist in germ cell specification[41], while perinuclear P granules in later stages of development are thought to help maintain germ cell identity by surveilling and processing nascent mRNAs as they exit the nucleus[42,43]. Furthermore, P granules are thought to maintain germ cell totipotency through the capture of nascent soma-specific transcripts and their asymmetric localisation has been proposed to promote microenvironment signalling events triggering asymmetric cell divisions (reviewed in[44]). In mammals, keratohyalin granules have been implicated in terminal differentiation of epidermal keratinocytes, through their dissolution in response to the acidic pH encountered once they reach the surface of the skin[45]. Accumulation of stress granules has also been observed during neuronal differentiation of human bone marrow-mesenchymal stem cells in response to the phosphorylation of eIF2α. Such condensates are required for the cell survival in response to the incubation to the differentiation media and are further dissolved to progress through the differentiation process[46].

Overall, our analysis of granule formation during differentiation has highlighted its differences to the assembly of granules associated with glucose starvation. Moreover, following the composition, assembly and position of distinct granules during differentiation over time has allowed us to describe the dynamics of stress granule formation and precisely correlate this with the stage of the development exhibited by the parasites. Notably this is quite distinct from the assembly and composition of granules under nutritional conditions conventionally used to analyse mRNP coalescence, revealing that the developmental response does not simply reflect the default response of trypanosomes to environmental stress. Rather the quorum-sensing driven response of the parasites generates a specific and programmed hierarchy of membraneless granule assembly, with the distinct components and their regulators such as TbDYRK contributing to the successful execution of the necessary steps for onward life cycle development.

## Methods

### Ethics statement
Animal experiments in this work were carried out in accordance with the local ethical approval requirements of the University of Edinburgh and the UK Home Office Animal (Scientific Procedures) Act (1986) under licence number PP2251183.

### Mouse infections
Female MF1 mice older than 8 weeks were used for all experiments and locally bred. They were kept at an ambient temperature of 21 °C and 56% humidity with 12 h of light and 12 h of dark cycle. Mice were inoculated with various *T. brucei* cell lines intraperitoneally (10,000 parasites) and the parasitaemia was monitored daily from day three post-infection. The appropriate life cycle forms of the parasite were harvested by collecting blood from trypanosome-infected mice by cardiac puncture. The parasites were then purified by separation on a diethyl aminoethyl cellulose (DEAE, pre-swollen DE52, Whatman) anion exchange column[47]. The parasites were subsequently counted using the Neubauer haemocytometer and then washed with Phosphate Buffered Saline-Glucose (PSG) (44 mM NaCl, 57 mM $Na_2HPO_4$, 3 mM $KH_2PO_4$; 55 mM glucose, pH 7.8) by centrifugation at $1000 \times g$ for 10 min. The cells were resuspended in an appropriate volume of HMI-9 supplemented with 10% v/v FBS at 37 °C in 5% CO2.

### Trypanosome culture, constructs and transfection
Pleomorphic *T. brucei* brucei EATRO 1125 AnTat1.1 J1339 (Parental) were cultured in vitro in HMI-9 supplemented with 10 % v/v FBS at 37 °C in 5 % CO2 as described in[28]. Slender forms were either grown in vitro or harvested from MF1 female mice at 3- or 4-days post infection (PI). Stumpy forms were harvested from MF1 female mice between 5- or 7-days PI.

Pleomorph transfections were performed as described by ref. [48] using Amaxa Nucleofector 2b with the programme Z-001 and in presence of the Tb-BSF transfection buffer (90 mM Sodium Phosphate pH 7.3, 5 mM KCl, 0.15 mM CaCl, 50 mM HEPES). Selection was applied by using the appropriate drugs: Hygromycin (HYG, 0.5 µg.mL⁻¹), Puromycin (PURO, 0.05 µg.mL⁻¹), Blasticidin (BSD, 2 µg.mL⁻¹) and Bleomycin (BLE, 0.8 µg.mL⁻¹).

CRISPR/Cas9 knock-out construct of TbDYRK (Tb927.10.15020) was generated as described in refs. [28,49] using the pPOTv6-BLA plasmid and transfected in the parental cell line. The endogenous add-back of the non-mutated (NM) or inactive version (S856G) version of the gene coding for the protein TbDYRK was generated by cloning the respective version into pPOTv7-HYG in place of the TYmNG tag as described in ref. [28]. This template was then used as described in ref. [49] to target one allele of the null mutant of TbDYRK by maintaining BSD and selecting with HYG. Correct sequences and insertions were assessed by PCR and sequencing (Primer list in Supplementary data 1).

Endogenous tagging (N-terminal or C-terminal) of proteins were generated as described in ref. [49] using either plasmids pPOTv6-BLA-TYmNG, pPOTv7-HYG-TYmNG[50] or modified versions of these plasmids pPOTv7-BLE-TYmNG, pPOTv6-BLA-Apex2-flag and pPOTv7-HYG-Apex2-flag (Primer list in Supplementary data 1).

*pPOT-vx-Apex2-flag.* 2xflag-Apex2-2xflag codon adapted sequence for trypanosomes, including the GS linkers sequences of pPOT plasmids, was synthesised by IDT and cloned into pPOT-vx_BLA or HYG using KpnI/SacI to replace the TYmNG sequence. Correct insertion was assessed by sequencing (Primer list in Supplementary data 1).

### Immunofluorescence
Cells from culture where wash twice with vPBS (137 mM NaCl, 3 mM KCl, 16 mM $Na_2HPO_4$, 3 mM $KH_2PO_4$, 46 mM sucrose) and fixed for 10 min with 4% v/v paraformaldehyde in vPBS. Permeabilization was performed by addition of 0.1% v/v Igepal CA-630 final for 10 min, followed by 10 min incubation with 0.1% m/v glycine in PBS. Cells were

then dried on poly-lysine treated slides. For cells coming from blood, the drop of blood was incubated with 500 μL of warm HMI-9 at 37 °C for 15 min then washed with vPBS. Parasites and blood cells were resuspended in 15 μL of vPBS then smeared onto poly-lysine treated slides, dried and ice-cold methanol fixed for 10 min. Blocking was performed with PBS 2% m/v BSA. Cell cycle analysis was carried out with 4',6-diamidino-2-phenylindole (DAPI) (100 ng.mL$^{-1}$) and primary and secondary antibodies were used as followed in PBS 0.2% m/v BSA: anti-PAD1 (1/1000)[51], anti-Ty BB2 (1/5)[52], anti-Flag (M2 monoclonal, Sigma-aldrich, 1/200), anti-Alba 3[15](1/800), goat anti-Rabbit Alexa 488 or 568 and goat anti-Mouse Alexa 488 or 568 (1/500). The slides were covered with coverslips after the mounting medium addition and analysed on a Zeiss Axio Imager Z2 mounted with a Prime BSI (Teledyne Photometrics) camera. μManager 2.0[53] was used for image capture and Fiji[54] for image analysis. PAD1 staining was acquired with settings enabling detection of low fluorescent cells. Such settings were maintained throughout the time course of the experiments to allow comparison between samples and time points, leading to signal saturation in high expression PAD1 cells. Measurement of PAD1 intensity was not used in these experiments. Particles analysis was performed using the following parameter: size = 0.12–1 μm and circularity = 0.7–1.

## Proximity labelling and pulldown

**Glucose starvation stress granules.** 2e$^7$ parasites were incubated for 35 min in 2 mL of complemented Creek's Minimal Medium (CMM) (9% v/v 10 kDa dialysed FBS, 1% v/v FBS, 0.1 mM of all amino-acids) complemented or not with 5.5 mM glucose. The 'no glucose' condition has been estimated to contain 0.15 mM glucose[55]. Based on an estimate of the normal glycolytic flux for BSF *T. brucei* 427 of 80 nmol.min$^{-1}$.(mg protein)$^{-1}$[56] and the assumption that one BSF trypanosome corresponds to 1.01e$^{-11}$ g protein[57], we approximated that a concentration of 1e$^7$ parasites per mL would have a glucose consumption rate of 8 nM.min$^{-1}$. With this approach we calculated that 0.15 mM glucose will be consumed in 18.75 min. 35 min was chosen as the starvation time point for proximity labelling approaches under these conditions to allow around 15 min of complete glucose starvation. One clone per cell line was used and replicates are from 3 independent cultures and proximity labelling experiments.

**Differentiation granules.** 0.6e$^7$ (early: parasitaemia <1e$^8$ / mL and PAD1 staining <1% of positive cells) or 2e$^7$ (late: parasitaemia > 2.5e$^8$/mL and PAD1 staining >40% of positive cells) parasites were purified from infected murine blood using DE52 anion exchange chromatography and incubated into 2 mL complemented CMM (9% v/v 10 kDa dialysed FBS, 1% v/v FBS, 0.1 mM of all amino-acids, 5.5 mM glucose) containing 1 mM biotin-tyramide for 30 min at 37 °C. One clone per cell line was used for infection and replicates are from three independent proximity labelling.

**Proximity labelling.** Cells were incubated for the time of the starvation/labelling with 1 mM biotin-tyramide. Fixation of the short-lived biotin radicals to proximal proteins was then performed by addition of 0.5 mM H$_2$O$_2$ for 1 min, then quenching was performed for 5 min with the addition of quenching buffer 2X (10 mM Trolox, 20 mM L-Ascorbic acid, pH7.3) in dPBS (137 mM NaCl, 2.7 mM KCl, 10 mM Na$_2$HPO$_4$, 1.8 mM KH$_2$PO$_4$, pH7.4). Samples were washed twice with quenching buffer 1X and once with dPBS. Lysis was performed with 500 μL of RIPA buffer 1X (RIPA buffer 10X: 1% v/v SDS, 5% m/v sodium deoxycholate, 10% v/v IgePal-CA-630, 1 mM EDTA, 1.25 M NaCl, 500 mM Tris pH 7.5, 1 tablet protease inhibitor (Roche) and 2 tablets PhosStop (Roche) per mL) containing 130 U benzonase (Merk). Samples were snap frozen, sonicated and lysates clarified by centrifugation.

**Pulldown.** Samples were incubated O/N at 6 °C with 50 μL beads (Dyneabeads™ MyOne™ Streptavidin C1, Invitrogen) on rotating

wheel. Elution was performed on magnetic rack, samples were washed 2 × 5 min with RIPA 1X and 2 × 5 min with Tris 50 mM pH 8.2 and eluted twice out of magnet with 0.1% Rapigest (Waters, UK) (in 50 mM Tris-HCl pH 8.2) at 50 °C for 10 min. 25 mM final DTT was then added and samples boiled 5 min at 95 °C, then cooled down before the addition of 8 M Urea. Samples were clarified by centrifugation then transferred to Vivaspin 500 spin column 10 K cartridge (Sartorius) and centrifuged. Columns were incubated in the dark with 0.05 M IAA (iodoacetamide - Sigma, Aldrich) in 8 M urea in 0.1 M Tris-HCl, pH 8.2 then washed with 8 M urea in 0.1 M Tris-HCl, pH 8.2, followed by incubation with 0.05 M ABC buffer (NH4HCO3 – Sigma-Aldrich) in ultrapure water. Protein digestion was performed with 1 μg trypsin/40 μg protein (Thermo Fisher) in 0.01% TFA, 0.05 M ABC buffer and incubated 14–16 h at 37 °C. Lys-c (1 μg to 50 μg proteins) was then added and incubate 2–4 h at 37 °C. Samples were washed with 0.05 mM ABC buffer and lysis stopped with the addition of 10% TFA until pH < 2.5. Peptides extracts were then cleaned on a C18 matrix stage tips pre-equilibrated (1: methanol, 2: 80% v/v ACN in 0.1% v/v TFA and 3: 0.1 % v/v TFA), and washed with 0.1% v/v TFA.

**Mass spectrometry.** Peptides were eluted in 40 μL of 80% acetonitrile in 0.1% TFA and concentrated down to 1 μL by vacuum centrifugation (Concentrator 5301, Eppendorf, UK). The peptide sample was then prepared for LC-MS/MS analysis by diluting it to 6 μL by 0.1% TFA.

LC-MS analyses were performed on an Orbitrap Fusion™ Lumos™ Tribrid™ Mass Spectrometer (Thermo Fisher Scientific, UK) both coupled on-line, to an Ultimate 3000 HPLC (Dionex, Thermo Fisher Scientific, UK). Peptides were separated on a 50 cm (2 μm particle size) EASY-Spray column (Thermo Scientific, UK), which was assembled on an EASY-Spray source (Thermo Scientific, UK) and operated constantly at 50ºC. Mobile phase A consisted of 0.1% formic acid in LC-MS grade water and mobile phase B consisted of 80% acetonitrile and 0.1% formic acid. Peptides were loaded onto the column at a flow rate of 0.3 μL min$^{-1}$ and eluted at a flow rate of 0.25 μL min$^{-1}$ according to the following gradient: 2 to 40% mobile phase B in 150 min and then to 95% in 11 min. Mobile phase B was retained at 95% for 5 min and returned to 2% a minute after until the end of the run (190 min).

MS1 scans were recorded at 120,000 resolution (scan range 350–1500 m/z) with an ion target of 4.0e5, and injection time of 50 ms. MS2 was performed in the ion trap at a rapid scan mode, with ion target of 2.0E4 and HCD fragmentation[58] with normalised collision energy of 27. The isolation window in the quadrupole was 1.4 Thomson. Only ions with charge between 2 and 6 were selected for MS2. Dynamic exclusion was set at 60 s.

The MaxQuant software platform[59] version 1.6.1.0 was used to process the raw files and search was conducted against the complete/reference proteome set of *Trypanosoma brucei brucei* (Uniprot database - released in April 2019), using the Andromeda search engine[60]. For the first search, peptide tolerance was set to 20 ppm while for the main search peptide tolerance was set to 4.5 pm. Isotope mass tolerance was 2 ppm and maximum charge to 7. Digestion mode was set to specific with trypsin allowing maximum of two missed cleavages. Carbamidomethylation of cysteine was set as fixed modification. Oxidation of methionine, phosphorylation of serine, threonine and tyrosine and biotinylation of lysine were set as variable modifications. Label-free quantitation analysis was performed by employing the MaxLFQ algorithm[61]. Absolute protein quantification was performed as previously described[62]. Peptide and protein identifications were filtered to 1% FDR.

The mass spectrometry proteomics data have been deposited to the ProteomeXchange Consortium via the PRIDE[63] partner repository with the dataset identifier PXD045841.

**Analysis.** Statistical enrichment analysis was performed using the bioconductor package DEP in R[64]. Proteins identified by peptides

present in at least 2 replicates of one condition were processed for further analysis. The data is background corrected and normalised by variance stabilising transformation. Missing values were then imputed using a 'Mixed' method: (1) missing at random values were imputed using the nearest neighbour averaging (knn) method and, (2) missing non-at random values were imputed using the imputation of left-censored missing data by random draws from a Gaussian distribution centred to a minimal value (MinProb) method. Protein-wise linear models, i.e. Moderated t-statistic, combined with empirical Bayes statistics, to moderate standard errors across genes, are used for the differential proximity analysis. Proteins presenting a fold change >1.5 and a $p < 0.05$ in a given condition were considered statistically enriched.

## Protein visualisation and western blotting
Protein samples were boiled for 5 min in Laemmli loading buffer, separated by SDS–PAGE (NuPAGE gel 4–12% Bis-Tris, Invitrogen) and blotted onto nitrocellulose membranes (Pierce). After blocking with TBS Odyssey Blocking buffer for at least 30 min at room temperature (RT), membranes were incubated with antibodies 1- to 3-h at RT or overnight at 4 °C under agitation in 2% BSA in 50% v/v Odyssey Blocking buffer / 50 % v/v TBS-T (0.1% v/v Tween in TBS). The primary antibodies were used at the following dilutions: anti-Flag (M2 monoclonal, Sigma-Aldrich, F1804 – 1/2000), anti-Alba 3[15] (1/2000); anti-Ty BB2 (1:5[52],); Streptavidin IRDye 680RD (1/1000, LI-COR). After 3 washes in TBS-T, proteins were visualised by incubating the membrane for 1 h at RT with a secondary antibody conjugated to a fluorescent dye diluted 1/5000 in 50% v/v Odyssey Blocking buffer/50% v/v TBS-T. Finally, membranes were scanned using a LI-COR Odyssey imager system.

## Reagents and biological resources
Additional reagents and biological resources are detailed in the Star method table in Supplementary Data 1.

## Bioinformatic tools, statistical analyses and data base referencing
*Trypanosoma spp* genes and protein sequences were retrieved from the web database TriTrypDB (http://tritrypdb.org/tritrypdb/)[65]. Small guide RNA (sgRNA), scaffold (G00) and template repair primers were designed using the programme at (www.leishgedit.net/Home.html) for the CRISPR-Cas9 gene manipulation[49]. Graphical and statistical analyses were carried out using Rstudio software (http://www.studio.org/) and R language (R Development Core Team (2005). R: A language and environment for statistical computing. R Foundation for Statistical Computing, Vienna, Austria. ISBN 3- 900051-07-0, URL: http://www.R-project.org). Further information can be found in the Star method table in supplement file 1. Data were examined before analysis to ensure normality and that no transformations were required. In case of non-normal distribution, the non-parametric Wilcoxon test was used to compare means. Network visualisation was performed using Cytoscape 3.10.1. When provided, boxplots represent the interquartile range (IQR) from the 1st (25th percentile, Q1) to the 3rd (75th percentile, Q3) quartile, the median and whiskers indicate the maximum (Q3 + 1.5*IQR) and minimum (Q1 − 1.5*IQR) values.

## Reporting summary
Further information on research design is available in the Nature Portfolio Reporting Summary linked to this article.

## Data availability
The raw proteomic data generated in this study have been deposited in the Pride database from the ProteomeXchange consortium with identifier PXD045841 [add hyp]. The processed proteomic data generated in this study are provided in the Supplementary Data file 2. The data generated from images analysis used in the Figs. 2, 4, 5B have been included in the supplementary data 1. Protein accession codes: ALBA4, Tb927.4.2030 ALBA3, Tb927.4.2040 XRNA, Tb927.7.4900 ZC3H20, Tb927.7.2660 PABP1, Tb927.9.9290 PABP2, Tb927.9.10770 RAB7, Tb927.9.11000 NEK17, Tb927.10.5950 DRC1, Tb927.10.7880 TbDYRK, Tb927.10.15020 ALBA2, Tb927.11.4450.

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

## Acknowledgements

We thank Guy Oldrieve for the modification of the pPOT plasmid to create the pPOTv7-BLEO-TYmNG. We also thank Philippe Bastin and Thierry Blisnick for providing the anti-Alba 3 antibody. Finally, we are very grateful to Paul Michels for his discussions, advice and proofing of the experiments provided in this manuscript. This work is funded by: a Wellcome Trust Investigator Award [103740/Z14/Z] to K.R.M., a Wolfson Research Merit Award [WM140045] to K.R.M., a Medical Research Council Career Development Award [MR/W026996/1] to M.C - this UK funded award is carried out in the frame of the Global Health EDCTP3 Joint Undertaking, a Wellcome Trust award [108504] to J.R. The funders had no role in the preparation of the manuscript.

## Author contributions

M.C., Conceptualisation, Data curation, Formal analysis, Investigation, Methodology, Supervision, Funding acquisition, Project administration; C.S., Data curation, Methodology; K.M.c.W., Methodology, Investigation; E.W., Investigation; J.R., Funding acquisition; K.R.M., Conceptualisation, Formal analysis, Supervision, Funding acquisition, Project administration.

## Competing interests

The authors declare no competing interests.
