## [Peer Review File · Nature Communications]

Differentiation granules, a dynamic regulator of *T. brucei* developmentREVIEWER COMMENTS

Reviewer #1 (Remarks to the Author):

During proliferation of *Trypanosoma brucei* in mammals, a quorum sensing mechanism triggers a differentiation from the 'slender' dividing form to the cell cycle arrested 'stumpy' form. There is little selective transcription of individual genes in trypanosomes and the regulation of gene expression during this transition is post transcriptional. This manuscript investigates the presence and composition of liquid liquid phase separation granules during the differentiation. The approach builds on a previous identification of a DYRK kinase as being necessary for differentiation to stumpy forms and the observation in other organisms that DYRK kinase affect granule formation. This is an interesting investigation that links granule composition to a cellular differentiation. The manuscript describes experiments that address and an interesting question.

The experiments take advantage of DYRK kinase null cell lines to investigate any function in granule formation. In the first experiment cells were starved on glucose and the sub cellular localisation of Alba 3, a known component of stress granules, determined by immunofluorescence using an antiserum that detected Alba3 and Alba 4 by western blotting. These experiments are good and show that the DRYK kinase is not necessary for starvation granule formation.

The next experiment investigated any role of DYRK in the dissolution of starvation granules after return of glucose to the medium. (The title of this section on line 117 is a little confusing). The cell lines used included a modified Alba4 locus expressing Alba4-Ty-mNG. The observations are presented as a quantitative measure of granule number (Figure 2A) and distribution that show differences in the dynamics of granule formation in DRYK -/- cells. I could not find: a statement of what parameters are used to define granule other than the Fiji software package, how many cells were counted to get to the percentage values for cells containing granules. A clearer explanation of the axes in figure 2B would be helpful.

Do granules form during differentiation? In these experiments cell lines containing either Alba3-Ty-mNG or Alba4-Ty-mNG were used to infect mice and trypanosomes isolated at days corresponding to predominantly slender or stumpy populations. The developmental stage was confirmed by immunofluorescence using anti-PAD1 a robust marker for stumpy forms. The sub cellular localisation of the Alba proteins was determined by anti-Ty immunofluorescence (Figure 3 and Figure supplement 2) and showed that both coalesce into granules as the cells arrest. Points of ambiguity with these experiments. It is not clear to me why immunofluorescence was used in these experiments (and those in figure 2 and others) when fluorescence from the mNG was available. Why was this experimental route chosen? Does the anti-Ty immunofluorescence show the same sub cellular distribution as the mNG? In Figure 3 there appears to be an increase in Alba3 expression over the time course, is this the case? Same for Alba 4.

There is a difference between starvation granules and those that appear in stumpy cells. This was shown by determining the subcellular localisation of PABP1 and 2 during differentiation to stumpy forms. It remains as a diffuse cytoplasmic and nuclear signal whereas PABP2 readily enters starvation granules. (No equivalent is shown for PABP1). This indicates a non-equivalence between the granules.

Next, a series of cell lines with different stress granule markers fused to -Ty-mNG were used to investigate the composition if the granules in stumpy form cells. A quantitative analysis of the presence or absence of these in granules in stumpy form cells is shown in Figure 4. These experiments provide further evidence for the composition of the granules and differences to stress granules.

The allocation of XRNA to STuRN requires a co-localisation experiment.

An analysis of granule formation showed that the DRYK kinase null cells had altered (Figure 5).

To analyse the composition of granules, cell lines, either wild type or DRYK kinase $-/-$, expressing - Flag-APEX2 fused to Alba3 or Alba4 were made for proximity biotinylation analysis. Biotinylation was then performed with and without glucose followed by streptavidin pull down and mass spec. Lines 256 257: were the repeats independent clones of APEX tagged cells?

Some commentary on the localisation of the identified proteins should be included given the wide range of distribution given.

The proximity biotinylation experiment would benefit from a more critical analysis of the data and integration with localisation data from tryptag. For example, XRND and XRNE (line 357) are normally localised to the nucleolus.

General points.

Were the various tagged cells lines validated? Does line 495/6 apply to all cell lines?

Some of the experiments would have benefited from co-localisation data

Some of the immunofluorescence images could be improved without further experimentation, for example the PAD1 immunofluorescence images. A matter of taste but black and white images for individual channels (as in Figure 1) are easier to see than colour (Figure 3 and beyond)

Reviewer #2 (Remarks to the Author):

In this paper, Cayla et al use proximity labeling and immunofluorescence assays to resolve the composition and distribution of stress granules formed in response to environmental conditions (glucose depletion) and during development (transition from slender to stumpy). In addition to scoring these characteristics in parental cells, the authors follow what happens to these processes when expression levels of the kinase DYRK involved in differentiation is reduced. The authors make the following conclusions from their data. First, they indicate that the composition of stress granules and differentiation granules are different. Second, they state that the kinase TbDYRK is not required for the initiation of differentiation granules but is involved in the resolution of granules during differentiation and the regulated spatial distribution of granules.

-The results are noteworthy and significant to the field as it highlights several areas of future research and does show that the differentiation response is different from the stress response initiated by glucose starvation. The data analysis and research design are sound and, provided the authors provide more details regarding their proximity labeling experiments, there is enough detail for the work to be reproduced. Some of the findings here differ from published work and this is discussed by the authors although I think their discussion may be an oversimplification. Please see comment 5 below. For the most part, the work supports the conclusions. The only issue I have is that I would like to see a more thorough discussion of the potential variation in the Alba3 and Alba4 proximity labeling experiments, especially as no validation experiments are included.

Comments and questions

1) I was unable to determine from the images in figure 1, particularly the TbDYRK $-/-$ panels, whether the conclusion reached by the authors that there was no difference in the number or size of Alba3-positive granules in parental or TbDYRK $-/-$ cell lines, was well supported. Additionally, the arrows highlighting granules were too small to interpret.

2) There should be more explanation regarding the data presentation in Figure 2B. I was unable to interpret the graph from the figure legend and materials and methods.

2) PAD1 staining in Figure 3 is saturated. While it is sufficient to resolve PAD1 positive from PAD 1 negative cells, could there be information that might be learned from having staining in the linear

range. For instance, does granule abundance/localization correlate with PAD1 expression levels?

3) Images in Figure 5A were of insufficient quality for me to judge the size, number, or localization of granules. On day 6 in parental cells, do Alba4 granules localize to nucleus?

4) In Figure 8, should there be Alba3, Alba 4 and PABP2 granules in the middle trypanosome? Also, Tb927.4.2920 is not mentioned in the figure legend. While it is discussed in supplemental figure 11, it would be helpful to discuss it specifically in this figure or eliminate it from the model. Are Alba3/4 and PABP2 in the same or different granules?

5) The Alba3 and Alba4 proximity proteomes generated in these experiments exhibit very little overlap with previous studies. The authors suggest that these differences are because of how the cells were treated (high density in 1.1% methyl cellulose vs in vivo development). I think that may be an oversimplification. What is the variability in these types of experiments? For example, how similar are the three replicates done here? If one were to repeat the experiment on different days, what would the variability look like? As I understand it, there were three replicates from a single differentiation experiment. While it is too expensive to repeat with biological and technical replicates, it is important to comment on variability on the assays for researchers that might do this in the future and for interpreting differences. This is especially important as there are no validation studies showing that proteins identified in the proximity labeling experiments colocalize with either Alba3 or 4.

6) Is it possible to do co-staining experiments with Alba3, Alba4 and other granules components to resolve if they are in the same granules? Alba 3 comes first and then Alba 4, but are they in the same granules or different ones? Is that possible to resolve?

7) Can the authors discuss how these finding relates to what is known about stress/differentiation granules in other organisms? If it is the first study of this kind, that is important.

Reviewer #3 (Remarks to the Author):

The manuscript by Caila and collaborators provides new and important data about the dynamics of RNA granules in trypanosomes. These parasites regulate their gene expression almost exclusively by post-transcriptional mechanisms mediated by the association of the transcripts with distinct sets of proteins. Most gene expression regulation occurs at the cytoplasm, where the mRNAs will be directed to the polyribosomes for translation or to storage granules to be stored or degraded according to the associated proteins. This dynamic association of mRNAs and RNA binding proteins also plays an essential role in the adaptation response of trypanosomes to the environment as they alternate between hosts during their life cycle and differentiate. Several publications in the last years have studied the interaction of distinct proteins to RNAs in RNA granules and provided evidence that the RNA-protein interactions are very dynamic. In this work, the group led by Keith Matthews provides new data about the dynamics and nature of RNA granules during the differentiation of African trypanosomes. They demonstrate the heterogeneity of the granules and how the composition and association of specific proteins change according to stress (glucose deprivation) and differentiation from slender to stumpy forms. The methodology is adequate and well employed to obtain the data and support the conclusions. The experimental protocols are well described, and the figures and captions are precise. This work is of great interest to those working with trypanosomes and cytoplasmic post-transcriptional gene expression regulation and should be accepted for publication.

We were very pleased that all three referees recognised the interest and importance of our study and are happy to address their comments on a point-by-point basis below.

Reviewer #1 (Remarks to the Author):

During proliferation of *Trypanosoma brucei* in mammals, a quorum sensing mechanism triggers a differentiation from the 'slender' dividing form to the cell cycle arrested 'stumpy' form. There is little selective transcription of individual genes in trypanosomes and the regulation of gene expression during this transition is post transcriptional. This manuscript investigates the presence and composition of liquid liquid phase separation granules during the differentiation. The approach builds on a previous identification of a DYRK kinase as being necessary for differentiation to stumpy forms and the observation in other organisms that DYRK kinase affect granule formation. This is an interesting investigation that links granule composition to a cellular differentiation. The manuscript describes experiments that address and an interesting question.

The experiments take advantage of DYRK kinase null cell lines to investigate any function in granule formation. In the first experiment cells were starved on glucose and the sub cellular localisation of Alba 3, a known component of stress granules, determined by immunofluorescence using an antiserum that detected Alba3 and Alba 4 by western blotting. These experiments are good and show that the DRYK kinase is not necessary for starvation granule formation.

The next experiment investigated any role of DYRK in the dissolution of starvation granules after return of glucose to the medium. (The title of this section on line 117 is a little confusing).

- Thank you - we agree and have modified the title to assist clarity (line 122, line numbering referring to the manuscript version including the tracked changes).

The cell lines used included a modified Alba4 locus expressing Alba4-Ty-mNG. The observations are presented as a quantitative measure of granule number (Figure 2A) and distribution that show differences in the dynamics of granule formation in DRYK $-/-$ cells. I could not find: a statement of what parameters are used to define granule other than the Fiji software package, how many cells were counted to get to the percentage values for cells containing granules. A clearer explanation of the axes in figure 2B would be helpful.

- We thank the reviewer for their comments and we have now added the missing information in the method and the figure legend (line 890-895). The number of cells counted has been added to the Supplement File 1.

Do granules form during differentiation? In these experiments cell lines containing either Alba3-Ty-mNG or Alba4-Ty-mNG were used to infect mice and trypanosomes isolated at days corresponding to predominantly slender or stumpy populations. The developmental stage was confirmed by immunofluorescence using anti-PAD1 a robust marker for stumpy forms. The sub cellular localisation of the Alba proteins was determined by anti-Ty immunofluorescence Figure 3 and Figure supplement 2) and showed that both coalesce into granules as the cells arrest. Points of ambiguity with these experiments. It is not clear to me why immunofluorescence was used in these experiments (and those in figure 2 and others) when fluorescence from the mNG was available. Why was this experimental route chosen? Does the anti-Ty immunofluorescence show the same sub cellular distribution as the mNG?

- Due to the nature of the experiments presented here, i.e. blood extraction of parasites from infected animals over a time series of several days we needed to use fixation methods that preserved the samples including ice cold methanol or 4% PFA. Unfortunately after such fixation the mNG signal is lost, explaining why we were required to use the anti-Ty antibody to detect the localisation of the tagged proteins by IFA. However we have used different tags such as Ty-mNG and Flag-APEX2 for Alba, for example, each of which provided the same

localisation in independent experiments. Furthermore, PABP1 and 2 presented a differential granular vs cytoplasmic localisation with the same Ty-mNG tag when assayed in starvation or differentiation conditions. In combination, this excludes a localisation artefact of the proteins when detected using the anti-Ty antibody as a possibility.

In Figure 3 there appears to be an increase in Alba3 expression over the time course, is this the case? Same for Alba 4.

- We thank the reviewer for this observation. Using the mean of fluorescence intensity per cell, as a proxy for the amount of Ty-tagged protein per cell, we have indeed observed that Alba3 and PABP1 accumulate when the cells are differentiating into stumpy form (PAD1+ cells). However this is not the case for the other markers we chose such as Alba 2/4, PABP2, XRNA, NEK17 or ZC3H20. We noted that the expression of PABP2 seems to present a transitory increase at D5 post infection. These results are now discussed in the text (lines 215-220) and the revised figure has been inserted in place of Figure Supplement 7.

There is a difference between starvation granules and those that appear in stumpy cells. This was shown by determining the subcellular localisation of PABP1 and 2 during differentiation to stumpy forms. It remains as a diffuse cytoplasmic and nuclear signal whereas PABP2 readily enters starvation granules. (No equivalent is shown for PABP1). This indicates a non-equivalence between the granules.

Next, a series of cell lines with different stress granule markers fused to -Ty-mNG were used to investigate the composition of the granules in stumpy form cells. A quantitative analysis of the presence or absence of these in granules in stumpy form cells is shown in Figure 4. These experiments provide further evidence for the composition of the granules and differences to stress granules. The allocation of XRNA to STuRN requires a co-localisation experiment.

- We have performed this experiment suggested by the reviewer using TbPIP39 as a marker of the STuRN. We observed that XRNA+ve granules are separated but proximal to the STuRN. We have now included these results in Figure supplement 5c and amended the text (line 231).

An analysis of granule formation showed that the DRYK kinase null cells had altered (Figure 5).

To analyse the composition of granules, cell lines, either wild type or DRYK kinase $-/-$, expressing -Flag-APEX2 fused to Alba3 or Alba4 were made for proximity biotinylation analysis. Biotinylation was then performed with and without glucose followed by streptavidin pull down and mass spec. Lines 256 257: were the repeats independent clones of APEX tagged cells?

- The replicates are of 3 independent cultures of the same APEX tagged clonal cell line. Further details have now been included in the text (line 581/588).

Some commentary on the localisation of the identified proteins should be included given the wide range of distribution given. The proximity biotinylation experiment would benefit from a more critical analysis of the data and integration with localisation data from tryptag. For example, XRND and XRNE (line 357) are normally localised to the nucleolus.

- Thank you for the suggestion. We have now integrated a discussion in the text (lines 345-353) and in figure supplement 13 presenting a network analysis focusing on the protein localisation. Interpretation of such analysis is complicated because no genome scale protein localisation has been performed to date on the relevant stumpy form. However, to minimise the effect of the life cycle stage we have used the LOPIT dataset described by Moloney et al, 20923 (PMID: 37479728) that was generated on the bloodstream form of the parasite rather than the Tryptag dataset that was generated from procyclic cells. There was inevitable complexity in the associated localisation of granule components but it was notable that there

were clear changes between early and later stage granules, this being made explicit in the revised text (lines 345-353).

Were the various tagged cells lines validated? Does line 495/6 apply to all cell lines?

- All cell lines were validated for the expression of the corresponding tagged proteins by western blots. As this is a routine procedure to validate the different cell lines in our pipeline, the blots also contain other proteins we are working on that are unrelated to this manuscript and have not been included here. However, such western blots can be provided upon request.

Some of the experiments would have benefited from co-localisation data

- We have now added colocalization analysis of PABP2 and Alba 3/4 during in vitro glucose starvation (Figure supplement 3a) as well as the colocalization of XRNA vs. PIP39 (Figure supplement 5c) during the time course of in vivo differentiation.

Some of the immunofluorescence images could be improved without further experimentation, for example the PAD1 immunofluorescence images.

- The PAD1 image saturation resulted because we used acquisition settings enabling the detection of D5 PAD1+ (low expression) and kept the same acquisition settings (to allow comparison) for the late time point where PAD1 signal was stronger. This allowed us to consistently discriminate between PAD1 positive and negative cells; we did not seek to quantitate the relative levels of PAD1 nor was this important in our assay - we simply wished to identify positive or negative cells. A comment has been added in the methods section to make this point (line 566-570).

A matter of taste but black and white images for individual channels (as in Figure 1) are easier to see than colour (Figure 3 and beyond)

- We thank the reviewer for this comment. In fact, the colour images we selected for the original submission were chosen to meet accepted inclusivity standards for colour-blind readers. However, we agree this does not generate the most visually attractive presentation. Therefore, we have now presented the main manuscript figure images using greyscale for the individual channels which we believe represent the images appropriately but remain accessible for all readers. The supplementary figures retain the original colour palette.

-

Reviewer #2 (Remarks to the Author):

In this paper, Cayla et al use proximity labeling and immunofluorescence assays to resolve the composition and distribution of stress granules formed in response to environmental conditions (glucose depletion) and during development (transition from slender to stumpy). In addition to scoring these characteristics in parental cells, the authors follow what happens to these processes when expression levels of the kinase DYRK involved in differentiation is reduced. The authors make the following conclusions from their data. First, they indicate that the composition of stress granules and differentiation granules are different. Second, they state that the kinase TbDYRK is not required for the initiation of differentiation granules but is involved in the resolution of granules during differentiation and the regulated spatial distribution of granules.

-The results are noteworthy and significant to the field as it highlights several areas of future research and does show that the differentiation response is different from the stress response initiated by glucose starvation. The data analysis and research design are sound and, provided the authors provide more details regarding their proximity labeling experiments, there is enough detail for the work to be reproduced. Some of the findings here differ from published work and this is discussed by the authors although I think their discussion may be an oversimplification. Please see comment 5 below. For the most part, the work supports the conclusions. The only issue I have is that I would like to see a more

thorough discussion of the potential variation in the Alba3 and Alba4 proximity labeling experiments, especially as no validation experiments are included.

Comments and questions

1) I was unable to determine from the images in figure 1, particularly the TbDYRK-/- panels, whether the conclusion reached by the authors that there was no difference in the number or size of Alba3-positive granules in parental or TbDYRK-/- cell lines, was well supported. Additionally, the arrows highlighting granules were too small to interpret.

- We thank the reviewer for this suggestion, however, with this experiment we were simply making the point that starvation stress granules can be formed in the absence of TbDYRK or the presence of a dominant negative mutant of TbDYRK. This conclusion was clear and we do not think detailed quantitation is required. Of course, further quantitation of the granules in the presence and absence of DYRK are available in Figure 2 and Figure 5.

2) There should be more explanation regarding the data presentation in Figure 2B. I was unable to interpret the graph from the figure legend and materials and methods.

- Further explanation have been added in the methods (line 893-896) and the figure legend as suggested by both reviewers 1 and 2.

2) PAD1 staining in Figure 3 is saturated. While it is sufficient to resolve PAD1 positive from PAD1 negative cells, could there be information that might be learned from having staining in the linear range. For instance, does granule abundance/localization correlate with PAD1 expression levels?

- We have replied to reviewer 1 for the same comment (see above). Briefly, due to our settings to detect low fluorescent PAD1+ cells we cannot use measurement of intensity of PAD1 expression but rather simply distinguish between positive and negative cells. We have added explanation for this point in the methods section (line 566-570).

3) Images in Figure 5A were of insufficient quality for me to judge the size, number, or localization of granules. On day 6 in parental cells, do Alba4 granules localize to nucleus?

- Figure 5 has been modified to increase the size of the images presented to facilitate visualisation. Alba 4 does not localise to the nucleus but rather remains peri-nuclear as we have also indicated in Figure 4B.

4) In Figure 8, should there be Alba3, Alba 4 and PABP2 granules in the middle trypanosome?

- Thank you for highlighting that our colour coding was difficult to interpret. The reviewer is right, Alba3/4 and PABP2 are in the middle parasite. We initially used the faded red colour for both PABPs and Alba3/4 to represent cytoplasmic diffuse signals that then condensate into the “red” starvation granules in the middle parasite. We have now modified the model to make things clearer. Furthermore, we have modified the model to represent the fact that Alba3/4 and PABP2 only partially colocalize in some granules (Figure supplement 3b).

Also, Tb927.4.2920 is not mentioned in the figure legend. While it is discussed in supplemental figure 11, it would be helpful to discuss it specifically in this figure or eliminate it from the model.

- We thank the reviewer for this observation and we agree that our interpretation was somewhat too speculative. Therefore we have now removed this figure, modified the text and model accordingly.

Are Alba3/4 and PABP2 in the same or different granules?

- We thank the reviewer for this comment and have now performed a complementary analysis where we colocalized PABP2 with Alba3/4 during an in vitro glucose starvation experiment (Figure supplement 3b, lines 180-182). Our results indicate that in presence of glucose a mainly diffuse signal is observed for both PABP2 and Alba3/4, with between 28.1 to 32.2%

of the pixels being shared in both channels. Using the non-completely specific antibody anti-Alba3/4 we observed some signal in a region reminiscent of the STURN that was not observed using the endogenously tagged Alba3 or Alba4 cell lines (Figure 3, Figure supplement 2). In the absence of Glucose, both PABP2 and Alba3/4 accumulate in granules but only between 15 to 19.6% are shared. These results indicate that PABP2 and Alba3/4 are mainly in different granules.

5) The Alba3 and Alba4 proximity proteomes generated in these experiments exhibit very little overlap with previous studies. The authors suggest that these differences are because of how the cells were treated (high density in 1.1% methyl cellulose vs in vivo development). I think that may be an oversimplification. What is the variability in these types of experiments? For example, how similar are the three replicates done here?

If one were to repeat the experiment on different days, what would the variability look like? As I understand it, there were three replicates from a single differentiation experiment. While it is too expensive to repeat with biological and technical replicates, it is important to comment on variability on the assays for researchers that might do this in the future and for interpreting differences. This is especially important as there are no validation studies showing that proteins identified in the proximity labeling experiments colocalize with either Alba3 or 4.

- We thank the reviewer for those very important comments and we have now added in the text the missing information (lines 281-284/316-319) as well as the PCA plots inserted in place of Figure Supplement 10 to present the variability in our experiments. These demonstrate a good reproducibility between replicates.

6) Is it possible to do co-staining experiments with Alba3, Alba4 and other granules components to resolve if they are in the same granules? Alba 3 comes first and then Alba 4, but are they in the same granules or different ones? Is that possible to resolve?

- We thank the reviewer for this suggestion. We have attempted to perform this analysis using the endogenously tagged Alba4::TYmNG and infected mice to reveal the colocalization with Alba 3 using the the anti-Alba3/4 antibody during in vivo differentiation. The idea was that if Alba3 and Alba4 would be in different granules, the antibody anti-Alba3/4 would reveal more granules than the antibody against the tag on Alba4. In our initial experiments we used the mouse IgG1 anti-TY antibody to reveal the endogenously tagged protein. However, the Anti-Alba3/4 antibody is a mouse sera and was incompatible with the anti-TY mouse monoclonal for colabelling experiments. As an alternative, we tried to use an Alpaca anti-mNeonGreen nanobody to detect the tagged Alba4::TYmNG. We were unsuccessful to confidently detect the tagged protein and therefore we were not able to perform the colocalization analysis of Alba3 and Alba4.

7) Can the authors discuss how these finding relates to what is known about stress/differentiation granules in other organisms? If it is the first study of this kind, that is important.

- Thank you for this suggestion. We have added Discussion on this point (line 482-497)

Reviewer #3 (Remarks to the Author):

The manuscript by Cayla and collaborators provides new and important data about the dynamics of RNA granules in trypanosomes. These parasites regulate their gene expression almost exclusively by post-transcriptional mechanisms mediated by the association of the transcripts with distinct sets of proteins. Most gene expression regulation occurs at the cytoplasm, where the mRNAs will be directed to the polyribosomes for translation or to storage granules to be stored or degraded according to the associated proteins. This dynamic association of mRNAs and RNA binding proteins also plays an essential role in the adaptation response of trypanosomes to the environment as they alternate between

hosts during their life cycle and differentiate. Several publications in the last years have studied the interaction of distinct proteins to RNAs in RNA granules and provided evidence that the RNA-protein interactions are very dynamic. In this work, the group led by Keith Matthews provides new data about the dynamics and nature of RNA granules during the differentiation of African trypanosomes. They demonstrate the heterogeneity of the granules and how the composition and association of specific proteins change according to stress (glucose deprivation) and differentiation from slender to stumpy forms. The methodology is adequate and well employed to obtain the data and support the conclusions. The experimental protocols are well described, and the figures and captions are precise. This work is of great interest to those working with trypanosomes and cytoplasmic post-transcriptional gene expression regulation and should be accepted for publication.

- We thank the reviewer for their very positive feedback.

REVIEWER COMMENTS

Reviewer #1 (Remarks to the Author):

The authors have answered all the points raised in my first report.

Reviewer #2 (Remarks to the Author):

Please find my responses to the authors rebuttal below. Original reviewer comments are in black font, authors rebuttal in red font and the reviewer's response to rebuttal in blue.

Reviewer: I was unable to determine from the images in figure 1, particularly the TbDYRK-/- panels, whether the conclusion reached by the authors that there was no difference in the number or size of Alba3-positive granules in parental or TbDYRK-/- cell lines, was well supported. Additionally, the arrows highlighting granules were too small to interpret.

Author rebuttal: We thank the reviewer for this suggestion, however, with this experiment we were simply making the point that starvation stress granules can be formed in the absence of TbDYRK or the presence of a dominant negative mutant of TbDYRK. This conclusion was clear and we do not think detailed quantitation is required. Of course, further quantitation of the granules in the presence and absence of DYRK are available in Figure 2 and Figure

Reviewer response: I am satisfied with this response about figure 1 but still have some concerns (described below) regarding figure 2 and the attempt to quantify the number and intensity of individual granules in a single parasite.

Reviewer: There should be more explanation regarding the data presentation in Figure 2B. I was unable to interpret the graph from the figure legend and materials and methods.

Author rebuttal: Further explanation have been added in the methods (line 893-896) and the figure legend as suggested by both reviewers 1 and 2.

Reviewer response: In the manuscript I have downloaded, lines 893-896 do not correspond to figure 2. I was unable to find the additional information describing figure 2 in material and methods of the revised manuscript. In the version I have, lines 144-149 (not 893-896) discuss figure 2B, which is the panel of primary concern. In the material and methods (as far as I could tell), lines 561 and 562 describe immunofluorescence but does not contain information about figure 2 and how mean fluorescence intensity was calculated. Also, in the legend for 2 (lines 715-728), there are many typos and grammatical errors. These should be corrected prior to resubmission. In figure 1, Alba antibodies exhibited diffuse staining with no granules indicated. However, the graph in Figure 2A, as I interpret it, indicates that most of the cells have granules. Am I not interpreting the data correctly? With the resolution of the images in Figures 1 and 2, the quantification of granule number and intensity is extremely challenging. As a result, it is difficult to resolve with any certainty small variations in granule number and intensities. Is this information essential to the paper? Could all of figure 2 be removed and the statements regarding a potential role for TbDYRK-/- in granule resolution tempered and perhaps removed. If the authors want to include this data, they should put more information into how the images were quantified. Currently, it only indicates (Lines 561-562) that FIJI was used and "particles analysis was performed using the following parameter: size.....". For example, per image, how many slices were analyzed (I am assuming these are confocal images and several optical images were gathered for each field). For fluorescence images of this low resolution, background is a significant problem. Lines 725-728 are unclear.

Reviewer Comment: PAD1 staining in Figure 3 is saturated. While it is sufficient to resolve PAD1

positive from PAD 1 negative cells, could there be information that might be learned from having staining in the linear range. For instance, does granule abundance/localization correlate with PAD1 expression levels?

Author rebuttal: We have replied to reviewer 1 for the same comment (see above). Briefly, due to our settings to detect low fluorescent PAD1+ cells we cannot use measurement of intensity of PAD1 expression but rather simply distinguish between positive and negative cells. We have added explanation for this point in the methods section (line 566-570).

Reviewer response: I understand and am satisfied with the explanation the authors provide for PAD1 saturation. However, lines 559-560 of the manuscript I have refers to Proximity labelling and pulldowns and not immunofluorescence experiments. Should this be corrected?

Reviewer Comment: Images in Figure 5A were of insufficient quality for me to judge the size, number, or localization of granules. On day 6 in parental cells, do Alba4 granules localize to nucleus?

Author rebuttal: Figure 5 has been modified to increase the size of the images presented to facilitate visualisation. Alba 4 does not localise to the nucleus but rather remains peri-nuclear as we have also indicated in Figure 4B.

Reviewer response: I am satisfied with the revised images in Figure 3.

Reviewer comment: In Figure 8, should there be Alba3, Alba 4 and PABP2 granules in the middle trypanosome?

Author rebuttal: Thank you for highlighting that our colour coding was difficult to interpret. The reviewer is right, Alba3/4 and PABP2 are in the middle parasite. We initially used the faded red colour for both PABPs and Alba3/4 to represent cytoplasmic diffuse signals that then condensate into the "red" starvation granules in the middle parasite. We have now modified the model to make things clearer. Furthermore, we have modified the model to represent the fact that Alba3/4 and PABP2 only partially colocalize in some granules (Figure supplement 3b).

Reviewer response: I am satisfied with the authors response and updates to the manuscript.

Reviewer comment: Also, Tb927.4.2920 is not mentioned in the figure legend. While it is discussed in supplemental figure 11, it would be helpful to discuss it specifically in this figure or eliminate it from the model.

Author response: We thank the reviewer for this observation, and we agree that our interpretation was somewhat too speculative. Therefore we have now removed this figure, modified the text and model accordingly.

Reviewer response: I am satisfied with the authors response and updates to the manuscript.

Reviewer comment: Are Alba3/4 and PABP2 in the same or different granules?

Author response: We thank the reviewer for this comment and have now performed a complementary analysis where we colocalized PABP2 with Alba3/4 during an in vitro glucose starvation experiment (Figure supplement 3b, lines 180-182). Our results indicate that in presence of glucose a mainly diffuse signal is observed for both PABP2 and Alba3/4, with between 28.1 to 32.2% of the pixels being shared in both channels. Using the non-completely specific antibody anti-Alba3/4 we observed some signal in a region reminiscent of the STURN that was not observed using the endogenously tagged Alba3 or Alba4 cell lines (Figure 3, Figure supplement 2). In the absence of Glucose, both PABP2 and Alba3/4 accumulate in granules but only between 15 to 19.6% are shared. These results indicate that PABP2 and Alba3/4 are mainly in different granules.

Reviewer response: I am satisfied with the authors response and updates to the manuscript.

Reviewer comments: The Alba3 and Alba4 proximity proteomes generated in these experiments exhibit very little overlap with previous studies. The authors suggest that these differences are because of how the cells were treated (high density in 1.1% methyl cellulose vs in vivo development). I think that may be an oversimplification. What is the variability in these types of experiments? For example, how similar are the three replicates done here? If one were to repeat the experiment on different days, what would the variability look like? As I understand it, there were three replicates from a single differentiation experiment. While it is too expensive to repeat with biological and technical replicates, it is important to comment on variability on the assays for researchers that might do this in the future and for interpreting differences. This is especially important as there are no validation studies showing that proteins identified in the proximity labeling experiments colocalize with either Alba3 or 4.

Author rebuttal: We thank the reviewer for those very important comments and we have now added in the text the missing information (lines 281-284/316-319) as well as the PCA plots inserted in place of Figure Supplement 10 to present the variability in our experiments. These demonstrate a good reproducibility between replicates.

Reviewer's response. I am satisfied with the authors response and updates to the manuscript.

Reviewer comment: Is it possible to do co-staining experiments with Alba3, Alba4 and other granules components to resolve if they are in the same granules? Alba 3 comes first and then Alba 4, but are they in the same granules or different ones? Is that possible to resolve?

Author rebuttal: We thank the reviewer for this suggestion. We have attempted to perform this analysis using the endogenously tagged Alba4::TYmNG and infected mice to reveal the colocalization with Alba 3 using the the anti-Alba3/4 antibody during in vivo differentiation. The idea was that if Alba3 and Alba4 would be in different granules, the antibody anti-Alba3/4 would reveal more granules than the antibody against the tag on Alba4. In our initial experiments we used the mouse IgG1 anti-TY antibody to reveal the endogenously tagged protein. However, the Anti-Alba3/4 antibody is a mouse sera and was incompatible with the anti-TY mouse monoclonal for colabelling experiments. As an alternative, we tried to use an Alpaca anti-mNeonGreen nanobody to detect the tagged Alba4::TYmNG. We were unsuccessful to confidently detect the tagged protein and therefore we were not able to perform the colocalization analysis of Alba3 and Alba4.

Reviewer response: I am satisfied with the authors response and updates to the manuscript.

Reviewer comment: Can the authors discuss how these finding relates to what is known about stress/differentiation granules in other organisms? If it is the first study of this kind, that is important.

Author rebuttal: Thank you for this suggestion. We have added Discussion on this point (line 482-497)

Reviewer's response:I am satisfied with the authors revisions.

New Reviewer comment on supplemental figure addition: In supplemental figure 10, do percentages given in the axes of the PCA plots refer to?

We thank the reviewers for their comments and interest in our study. We have only replied in the following response where there were remaining queries.

REVIEWER COMMENTS

Reviewer #1 (Remarks to the Author):

The authors have answered all the points raised in my first report.

Reviewer #2 (Remarks to the Author):

Please find my responses to the authors rebuttal below. Original reviewer comments are in black font, authors rebuttal in red font and the reviewer's response to rebuttal in blue.

Reviewer: I was unable to determine from the images in figure 1, particularly the TbDYRK-/- panels, whether the conclusion reached by the authors that there was no difference in the number or size of Alba3-positive granules in parental or TbDYRK-/- cell lines, was well supported. Additionally, the arrows highlighting granules were too small to interpret.

Author rebuttal: We thank the reviewer for this suggestion, however, with this experiment we were simply making the point that starvation stress granules can be formed in the absence of TbDYRK or the presence of a dominant negative mutant of TbDYRK. This conclusion was clear and we do not think detailed quantitation is required. Of course, further quantitation of the granules in the presence and absence of DYRK are available in Figure 2 and Figure

Reviewer response: I am satisfied with this response about figure 1 but still have some concerns (described below) regarding figure 2 and the attempt to quantify the number and intensity of individual granules in a single parasite.

Reviewer: There should be more explanation regarding the data presentation in Figure 2B. I was unable to interpret the graph from the figure legend and materials and methods.

Author rebuttal: Further explanation have been added in the methods (line 893-896) and the figure legend as suggested by both reviewers 1 and 2.

Reviewer response: In the manuscript I have downloaded, lines 893-896 do not correspond to figure 2. I was unable to find the additional information describing figure 2 in material and methods of the revised manuscript. In the version I have, lines 144-149 (not 893-896)

discuss figure 2B, which is the panel of primary concern. In the material and methods (as far as I could tell), lines 561 and 562 describe immunofluorescence but does not contain information about figure 2 and how mean fluorescence intensity was calculated. Also, in the legend for 2 (lines 715-728), there are many typos and grammatical errors. These should be corrected prior to resubmission. In figure 1, Alba antibodies exhibited diffuse staining with no granules indicated. However, the graph in Figure 2A, as I interpret it, indicates that most of the cells have granules. Am I not interpreting the data correctly? With the resolution of the images in Figures 1 and 2, the quantification of granule number and intensity is extremely challenging. As a result, it is difficult to resolve with any certainty small variations in granule number and intensities. Is this information essential to the paper? Could all of figure 2 be removed and the statements regarding a potential role for TbDYRK-/- in granule resolution tempered and perhaps removed. If the authors want to include this data, they should put more information into how the images were quantified. Currently, it only indicates (Lines 561-562) that FIJI was used and "particles analysis was performed using the following parameter: size.....". For example, per image, how many slices were analyzed (I am assuming these are confocal images and several optical images were gathered for each field). For fluorescence images of this low resolution, background is a significant problem. Lines 725-728 are unclear.

- Author response: We apologise for the line numbering confusion. As stated in our first response to the reviewers, we used line numbering referring to the version where the tracked changes were visible, while the reviewers appear to have been provided with a different version. To avoid further confusion, we are now using line numbering referring to a version where tracked changes are not visible.

Regarding figure 2b, as indicated in our response to the editors, we understand that this figure remained challenging to understand and we have decided to remove it from the manuscript. Indeed, as highlighted by the reviewer, the information provided by this figure is not required to the understanding of the study. Accordingly, we have removed our interpretation that granules in the TbDYRK-/- could be denser (Lines 144-146; 376 and 720-722). However, removing this dataset does not affect our conclusions and discussions. We have maintained figure 2a as it provides the quantification of the number of granules in the presence or absence of glucose. To quantify the number of granules we used the FIJI particle analysis tool, using the parameters provided in the 'materials and methods', providing an unbiased quantification method. This indicates, indeed, that there is a small basal number of granules identifiable in culture in the +glucose condition and that this number strongly increases in absence of glucose, to finally return quickly to the basal level when glucose is repleted. This is affected by the absence of TbDYRK where less variation can be observed. All this discussion is already present in the main manuscript.

Reviewer Comment: PAD1 staining in Figure 3 is saturated. While it is sufficient to resolve PAD1 positive from PAD 1 negative cells, could there be information that might be learned from having staining in the linear range. For instance, does granule abundance/localization correlate with PAD1 expression levels?

Author rebuttal: We have replied to reviewer 1 for the same comment (see above). Briefly, due to our settings to detect low fluorescent PAD1+ cells we cannot use measurement of intensity of PAD1 expression but rather simply distinguish between positive and negative cells. We have added explanation for this point in the methods section (line 566-570).

Reviewer response: I understand and am satisfied with the explanation the authors provide for PAD1 saturation. However, lines 559-560 of the manuscript I have refers to Proximity labelling and pulldowns and not immunofluorescence experiments. Should this be corrected?

- Author response: As we indicated before, line numbering referred to a version perhaps unavailable to the reviewer. The text provided in the manuscript is accurate and can be found in Lines 556-559, while the "Proximity labelling and pulldowns" section is found from line 561.

Reviewer Comment: Images in Figure 5A were of insufficient quality for me to judge the size, number, or localization of granules. On day 6 in parental cells, do Alba4 granules localize to nucleus?

Author rebuttal: Figure 5 has been modified to increase the size of the images presented to facilitate visualisation. Alba 4 does not localise to the nucleus but rather remains perinuclear as we have also indicated in Figure 4B.

Reviewer response: I am satisfied with the revised images in Figure 3.

Reviewer comment: In Figure 8, should there be Alba3, Alba 4 and PABP2 granules in the middle trypanosome?

Author rebuttal: Thank you for highlighting that our colour coding was difficult to interpret. The reviewer is right, Alba3/4 and PABP2 are in the middle parasite. We initially used the faded red colour for both PABPs and Alba3/4 to represent cytoplasmic diffuse signals that then condensate into the "red" starvation granules in the middle parasite. We have now modified the model to make things clearer. Furthermore, we have modified the model to represent the fact that Alba3/4 and PABP2 only partially colocalize in some granules (Figure supplement 3b).

Reviewer response: I am satisfied with the authors response and updates to the manuscript.

Reviewer comment: Also, Tb927.4.2920 is not mentioned in the figure legend. While it is discussed in supplemental figure 11, it would be helpful to discuss it specifically in this figure or eliminate it from the model.

Author response: We thank the reviewer for this observation, and we agree that our interpretation was somewhat too speculative. Therefore we have now removed this figure, modified the text and model accordingly.

Reviewer response: I am satisfied with the authors response and updates to the manuscript.

Reviewer comment: Are Alba3/4 and PABP2 in the same or different granules?

Author response: We thank the reviewer for this comment and have now performed a complementary analysis where we colocalized PABP2 with Alba3/4 during an in vitro glucose starvation experiment (Figure supplement 3b, lines 180-182). Our results indicate that in presence of glucose a mainly diffuse signal is observed for both PABP2 and Alba3/4, with between 28.1 to 32.2% of the pixels being shared in both channels. Using the non-completely specific antibody anti-Alba3/4 we observed some signal in a region reminiscent of the STURN that was not observed using the endogenously tagged Alba3 or Alba4 cell lines (Figure 3, Figure supplement 2). In the absence of Glucose, both PABP2 and Alba3/4 accumulate in granules but only between 15 to 19.6% are shared. These results indicate that PABP2 and Alba3/4 are mainly in different granules.

Reviewer response: I am satisfied with the authors response and updates to the manuscript.

Reviewer comments: The Alba3 and Alba4 proximity proteomes generated in these experiments exhibit very little overlap with previous studies. The authors suggest that these differences are because of how the cells were treated (high density in 1.1% methyl cellulose vs in vivo development). I think that may be an oversimplification. What is the variability in these types of experiments? For example, how similar are the three replicates done here? If one were to repeat the experiment on different days, what would the variability look like? As I understand it, there were three replicates from a single differentiation experiment. While it is too expensive to repeat with biological and technical replicates, it is important to comment on variability on the assays for researchers that might do this in the future and for interpreting differences. This is especially important as there are no validation studies showing that proteins identified in the proximity labeling experiments colocalize with either Alba3 or 4.

Author rebuttal: We thank the reviewer for those very important comments and we have now added in the text the missing information (lines 281-284/316-319) as well as the PCA plots inserted in place of Figure Supplement 10 to present the variability in our experiments. These demonstrate a good reproducibility between replicates.

Reviewer's response. I am satisfied with the authors response and updates to the manuscript.

Reviewer comment: Is it possible to do co-staining experiments with Alba3, Alba4 and other granules components to resolve if they are in the same granules? Alba 3 comes first and then Alba 4, but are they in the same granules or different ones? Is that possible to resolve?

Author rebuttal: We thank the reviewer for this suggestion. We have attempted to perform this analysis using the endogenously tagged Alba4::TYmNG and infected mice to reveal the colocalization with Alba 3 using the the anti-Alba3/4 antibody during in vivo differentiation. The idea was that if Alba3 and Alba4 would be in different granules, the antibody anti-Alba3/4 would reveal more granules than the antibody against the tag on Alba4. In our initial experiments we used the mouse IgG1 anti-TY antibody to reveal the endogenously tagged protein. However, the Anti-Alba3/4 antibody is a mouse sera and was incompatible with the anti-TY mouse monoclonal for colabelling experiments. As an alternative, we tried to use an Alpaca anti-mNeonGreen nanobody to detect the tagged Alba4::TYmNG. We were unsuccessful to confidently detect the tagged protein and therefore we were not able to perform the colocalization analysis of Alba3 and Alba4.

Reviewer response: I am satisfied with the authors response and updates to the manuscript.

Reviewer comment: Can the authors discuss how these finding relates to what is known about stress/differentiation granules in other organisms? If it is the first study of this kind, that is important.

Author rebuttal: Thank you for this suggestion. We have added Discussion on this point (line 482-497)

Reviewer's response:I am satisfied with the authors revisions.

New Reviewer comment on supplemental figure addition: In supplemental figure 10, do percentages given in the axes of the PCA plots refer to?

- Author response: We are not sure we fully understand the question and whether the reviewer is asking what is the meaning of the principal components (PCs) in the principal component analysis or not? If it is the case we would refer to the relevant literature explaining how such PCs are calculated. Nevertheless, we have added text to the figure legend (lines 881-884) clarifying that the percentages represent the principal components (PCs) explaining the variance of the data analysed. 2 PCs are indicated in our graphs, PC1 on the x-axis and PC2 on the y-axis. These represent the 2 PCs that explain most of the variance of the data. Indeed, together they explain 32.3 and 60.1 % of the variance of the starvation and differentiation proximity labelling datasets, respectively.